# Cytokines regulate the antigen-presenting characteristics of human circulating and tissue-resident intestinal ILCs

Anna Rao[1✉], Otto Strauss[1,7], Efthymia Kokkinou[1,7], Mélanie Bruchard[2], Kumar P. Tripathi[3], Heinrich Schlums[4], Anna Carrasco[1], Luca Mazzurana[1], Viktoria Konya[1], Eduardo J. Villablanca [3], Niklas K. Björkström [1], Ulrik Lindforss[5], Hergen Spits [2] & Jenny Mjösberg[1,6✉]

ILCs and T helper cells have been shown to exert bi-directional regulation in mice. However, how crosstalk between ILCs and CD4+ T cells influences immune function in humans is unknown. Here we show that human intestinal ILCs co-localize with T cells in healthy and colorectal cancer tissue and display elevated HLA-DR expression in tumor and tumor-adjacent areas. Although mostly lacking co-stimulatory molecules ex vivo, intestinal and peripheral blood (PB) ILCs acquire antigen-presenting characteristics triggered by inflammasome-associated cytokines IL-1β and IL-18. IL-1β drives the expression of HLA-DR and co-stimulatory molecules on PB ILCs in an NF-κB-dependent manner, priming them as efficient inducers of cytomegalovirus-specific memory CD4+ T-cell responses. This effect is strongly inhibited by the anti-inflammatory cytokine TGF-β. Our results suggest that circulating and tissue-resident ILCs have the intrinsic capacity to respond to the immediate cytokine milieu and regulate local CD4+ T-cell responses, with potential implications for anti-tumor immunity and inflammation.

[1] Center for Infectious Medicine, Department of Medicine, Karolinska Institutet, Karolinska University Hospital Huddinge, Stockholm, Sweden. [2] Department of Experimental Immunology, Academic Medical Center at the University of Amsterdam, Amsterdam, Netherlands. [3] Immunology and Allergy Division, Department of Medicine, Solna, Karolinska Institutet and Karolinska University Hospital, Stockholm 17164, Sweden. [4] Center for Hematology and Regenerative Medicine, Department of Medicine, Karolinska Institutet, Stockholm, Sweden. [5] Department of Molecular Medicine and Surgery, Karolinska Institutet and Center for Digestive Diseases, Karolinska University Hospital, Stockholm, Sweden. [6] Department of Clinical and Experimental Medicine, Linköping University, Linköping, Sweden. [7] These authors contributed equally: Otto Strauss, Efthymia Kokkinou. ✉email: anna.rao@ki.se; jenny.mjosberg@ki.se

The ability of innate lymphoid cells (ILCs) to sense and amplify inflammatory signals allows their interaction with an array of immune cells. Based on the expression of master transcription factors and effector cytokines ILCs are classically divided into three major groups: ILC1, ILC2, and ILC3, which function as the innate counterparts of $T_H1$, $T_H2$, and $T_H17/22$ cells, respectively[1]. ILC1 produce IFN-γ and TNF and depend on the transcription factor T-bet for their development[1]. During intestinal inflammation ILC1 accumulate in the inflamed mucosa, as previously shown in patients with Crohn's disease[2]. ILC2 are characterized by the expression of the transcription factor GATA3[3], and produce IL-13, IL-5, IL-4, and IL-9 in response to IL-33, IL-25, and TSLP stimulation[4–6]. In particular, human ILC2 are identified by CRTH2 and CD161 surface expression[7] and can be found in a variety of mucosal and non-mucosal tissues including blood[7–9]. ILC3 are dependent on RORγT expression for their development and function[10], and in the human, they are characterized by CD117 expression and production of IL-22, IL-17F, and GM-CSF[11–14]. Judging by surface marker expression, all three ILC subsets can be found in peripheral circulation of healthy individuals. However, cells that phenotypically resemble ILC3 (ILC3-like cells) in blood are in fact a mixture of immature ILC precursors (ILCP)[15,16]. Circulating ILCP can give rise to all ILC subsets in vitro and in vivo, demonstrating major differences between fully differentiated tissue-associated ILC2 and ILC3 and ILC2 and ILC3-like ILCP in the blood.

Using single-cell RNA sequencing we demonstrated the presence of a transcriptionally distinct population of ILC3 in human tonsils expressing *HLA-DR* and *HLA-DQ*[14]. ILC3 expressing *HLA-DR/DQ* were enriched for transcripts encoding *CD74* (HLA-DR invariant chain) and *CTSS* (encoding cathepsin S), implying that they may possess the capacity to present antigens. MHCII-mediated crosstalk between ILCs and T cells has been demonstrated in several studies of genetically-engineered mice[17–22]. ILC3 have been shown to either stimulate[19,23] or suppress[20,21] T-cell activity, depending on the nature of the interaction. During intestinal homeostasis, mouse MHCII+ ILC3 were shown to contribute to immune tolerance by depletion of commensal bacteria-specific T cells[20,21]. Conversely, in a model of acute colitis, TNF-like ligand 1A (TL1A)-activated intestinal ILC3 were shown to stimulate antigen-specific T cells[23]. Similarly, under the influence of IL-1β, peripheral mouse NKp46− ILC3 upregulate MHCII and co-stimulatory molecules, allowing them to prime naive CD4+ T cells and induce their proliferation[19]. The capacity of HLA-DR+ ILCs to regulate T-cell responses in humans remains elusive. Since ILCs are particularly accumulated at mucosal sites[24,25], where naive T cells are scarce, we set out to determine the role for human HLA-DR+ ILCs in regulating memory CD4+ T-cell responses in inflammation or under steady-state conditions.

Here we show that ILCs in colorectal tumors display elevated HLA-DR expression and frequently co-localize with T cells in situ. Furthermore, we address potential cytokine networks involved in regulating the antigen-presenting properties of human ILCs in colorectal cancer (CRC). Exposure of peripheral blood (PB) and intestinal ILCs to IL-1β or IL-18 leads to upregulation of HLA-DR and induction of co-stimulatory molecules. For PB ILCs, the antigen-presenting characteristics induced by IL-1β are dependent on NF-κB. IL-1β promotes the ability of PB ILCs to induce autologous cytomegalovirus (CMV)-specific memory CD4+ T-cell responses, demonstrating the functional capacity of ILCs for antigen uptake, processing and presentation. These properties are efficiently counteracted by TGF-β in PB ILC3-like cells. Better understanding of ILC-T-cell interactions and how they depend on the immediate cytokine

microenvironment could be harnessed for improved immunomodulatory treatments.

## Results

**CRC ILCs have increased HLA-DR and co-localize with T cells.** We previously demonstrated the presence of a transcriptionally distinct HLA-DR+ CD127+ ILC3 subset in human tonsil based on single-cell RNA sequencing[14]. Here, we investigated whether a phenotypically similar population can be detected in non-affected and/or diseased human colon of CRC patients. ILCs from three sub-anatomical regions in the colon: non-affected tissue, tumor border and central tumor tissue were analyzed for HLA-DR expression by flow cytometry (Fig. 1a, b; Supplementary Fig. 1a-d). Although all regions showed similar ILC frequencies (Supplementary Fig. 1e), increased HLA-DR expression, in terms of percentage and mean fluorescence intensity (MFI), was detected on ILCs at the border of colorectal tumors (Fig. 1a, b; Supplementary Fig. 1c, d). A similar increase in MFI was seen in the center of the tumor mass. HLA-DR upregulation on ILCs was not clearly correlated to the cancer stage (Fig. 1b) but it was confined to the intestine, as we did not observe any differences in HLA-DR expression on PB ILCs between healthy donors and patients with CRC (Supplementary Fig. 1f). HLA-DR was upregulated on both the CD117+ and CD117− ILC subsets in the tumor (Fig. 1a; Supplementary Fig. 1c, d), while HLA-DR and CRTH2 expression, the latter marking human ILC2[7], was mutually exclusive (Supplementary Fig. 1b).

In contrast to PB ILCs, most ILCs in non-affected colonic tissue expressed CD69, a marker for tissue residency[26], together with ILC3 markers NKp44 and RORγT[14,27] (Supplementary Figs. 1g and 2a, b). The expression of CD69, NKp44 and RORγT was significantly reduced in the tumor-associated ILCs, indicating possible infiltration of circulating cells into the tumor tissue. Nonetheless, HLA-DR was detected on both CD69+ and CD69− ILCs in the tumor (Supplementary Fig. 1h). Although the expression of RORγT in ILCs gradually declined toward the center of the tumor mass, there was no concordant upregulation of T-bet, ruling out generation of ILC1 from ILC3 as previously reported in Crohn's disease[2] (Supplementary Fig. 2a, b). In summary, tumor-associated ILCs exhibit upregulated HLA-DR expression and display a surface phenotype indicative of infiltration of circulating cells.

We next sought to investigate the localization of HLA-DR expressing ILCs and their proximity to T cells in situ. For this we performed multicolor immunofluorescence confocal microscopy of non-affected colon tissue as well as the tumor border. Due to their heterogeneous expression of RORγT (Supplementary Fig. 2a, b) and CD117 (Fig. 1a) colonic ILCs were identified as CD127+CD3−CD45+ lymphocytes. Ruling out significant contamination by other cell types, ex vivo isolated CD3−CD127+ lymphocytes from colon were largely negative for lineage markers and, as previously reported[28], contained only a minor proportion of CD94+ cells (Supplementary Fig. 2c). ILCs were detected in both non-affected colon (Fig. 1c, Supplementary Fig. 3b) and at the border of the tumor (Fig. 1d, Supplementary Fig. 4b), where they localized either around the crypts together with HLA-DR^hiCD45+ cells or in the lymphoid follicles. When specifically assessing HLA-DR-expressing ILCs, these were found either isolated or in co-localization with T cells in both anatomical regions (Fig. 1c, d; Supplementary Figs. 3b, 4b). Quantification of cellular co-localizations revealed that on average 52% and 38% of total ILCs were found in the immediate vicinity of a T cell in the non-affected or tumor border area, respectively, either alone or together with an HLA-DR^hiCD45+ cell (Fig. 1e). Approximately one-third of ILCs co-localized with an

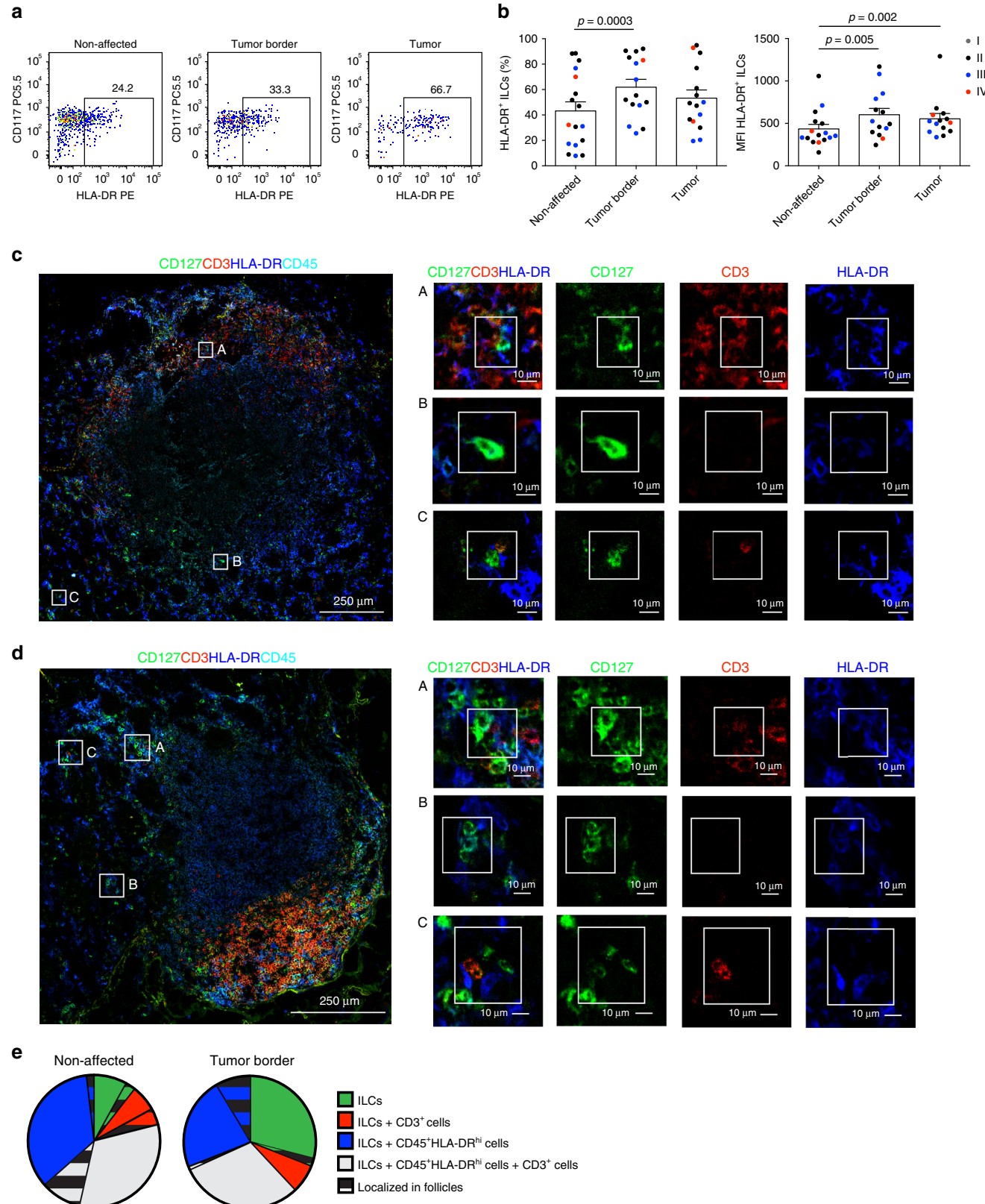

HLA-DR$^{hi}$CD45$^+$ cell, both in the non-affected area and at the border of the tumor.

Importantly, despite the expression of HLA-DR, we did not detect T-cell co-stimulatory or co-inhibitory molecules on intestinal ILCs from the majority of analyzed patients (10 out of 13), independent of their sub-anatomical origin (Supplementary Fig. 5). In three CRC patients, low frequencies of CD86 expression on tumor-associated ILCs were observed (Supplementary Fig. 6), suggesting that ILCs may, under certain circumstances, acquire antigen-presenting characteristics in vivo.

**Fig. 1 HLA-DR expressing ILCs in non-affected colon and colorectal cancer tissue. a** Representative flow cytometric dot plots of HLA-DR and CD117 expression as well as **b** frequency and MFI of HLA-DR expression on ILCs from non-affected colon, tumor border and central tumor tissues. Individual data points are color-coded based on the cancer stage of the donor. $N$ (patients) = 17 (frequency) and 16 (MFI). Not all sub-anatomical regions could be obtained for every donor; source data are provided as a source data file. Bars and error bars indicate mean and SEM; statistical significance was assessed using two-sided Wilcoxon matched-pairs signed rank test. **c, d** Multicolor immunofluorescence microscopy demonstrating distribution and HLA-DR expression of CD127+CD3−CD45+ ILCs and CD3+ T cells in (**c**) non-affected colon and (**d**) at the border of colorectal tumors. Magnified regions depict examples of (A) HLA-DR+CD127+ ILCs in contact with CD3+ T cells, (B) HLA-DR+CD127+ ILCs alone, and (C) CD127+ ILCs co-localized with HLA-DRhiCD45+ cells and CD3+ T cells; depicted are lymphoid follicle-containing regions, representative of three patients analyzed in three independent experiments. **e** Quantification of cellular co-localizations in the non-affected and tumor border colon tissue. Depicted is the proportion of ILCs co-localizing with CD3+CD45+ and/or CD45+HLA-DRhi cells inside and outside of lymphoid follicles. Source data are provided as a source data file.

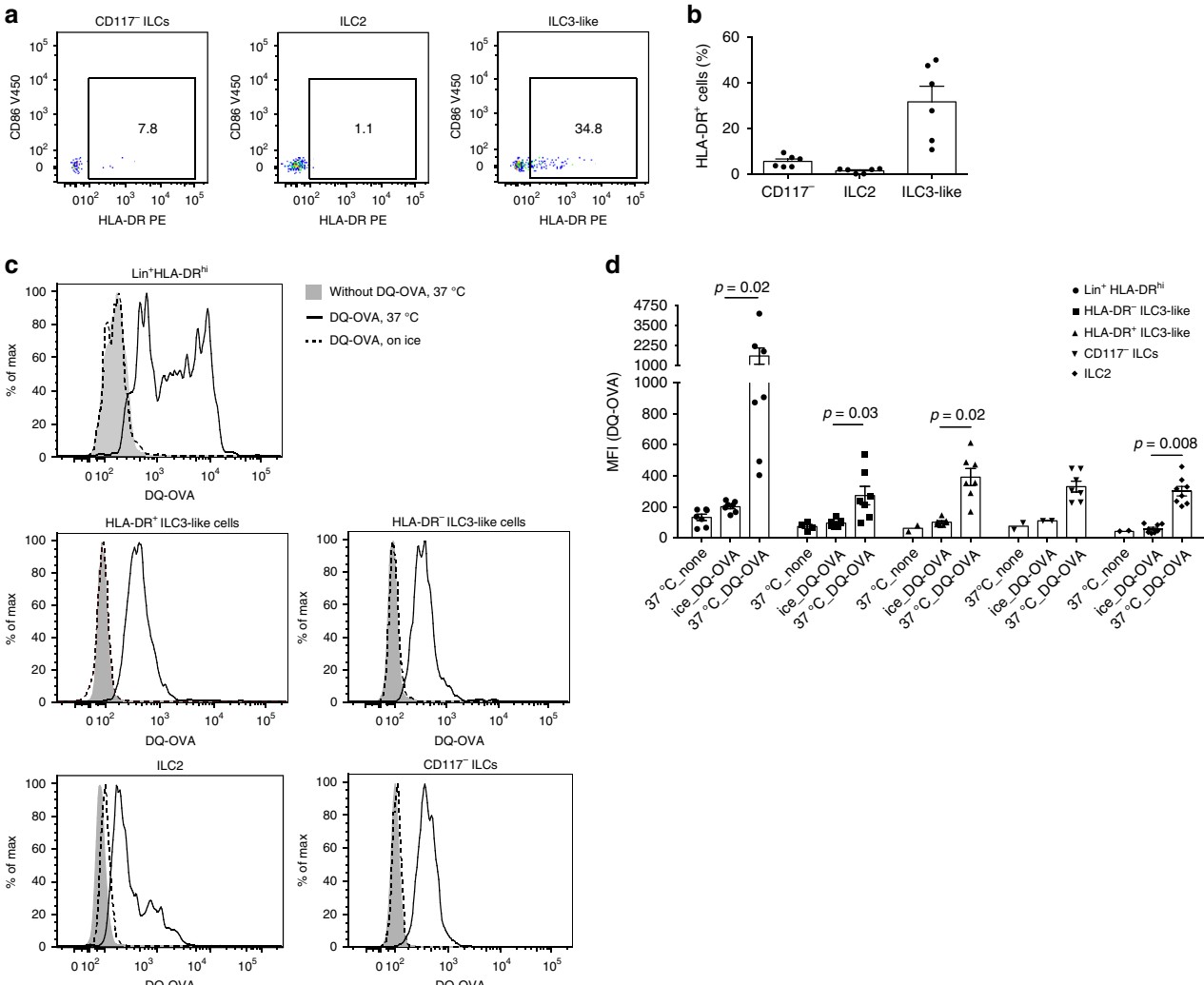

**Fig. 2 HLA-DR and co-stimulatory molecule expression on and DQ-OVA uptake by PB ILC subsets. a** Representative flow cytometric dot plots of HLA-DR and CD86 expression on PB CD117−ILCs, ILC2 and ILC3-like cells, defined as Lin−CD127+CD161+CRTH2−CD117−, Lin−CD127+CD161+CRTH2+ and Lin−CD127+CD161+CRTH2−CD117+ cells, respectively. **b** Mean of HLA-DR expression on PB ILC subsets. $N$ (donors) = 6; bars indicate mean value; error bars indicate SEM. **c** Representative histograms and **d** summarized data of fluorescence emission by sort-purified PB CD117− ILCs, ILC2, HLA-DR+ ILC3-like cells, HLA-DR− ILC3-like cells and Lin+HLA-DRhi cells after 4 h incubation with 10 μg/ml DQ-OVA. **c** Gray filled histograms depict cells incubated without DQ-OVA, solid black lines show cells incubated with DQ-OVA at 37 °C, and dashed black lines—cells incubated with DQ-OVA on ice. **d** $N$ (donors) = 7. Due to limited cell numbers, we could not generate data for all conditions for every donor. Bars and error bars indicate mean and SEM; statistical significance was assessed using two-sided Wilcoxon matched-pairs signed rank test. Source data are provided as a source data file.

**PB ILCs are able to take up and process full protein.** HLA-DR+ ILCs were also readily detectable in PB of healthy individuals (Fig. 2a, b). On average 5% of the CD117− ILCs and 32% of the ILC3-like cells expressed HLA-DR (Fig. 2b). ILC2 did not display noticeable HLA-DR expression. PB ILCs did not express T-cell co-stimulatory or co-inhibitory molecules (Supplementary Fig. 5).

Given that HLA-DR+ ILCs were detected in both blood and gut and the majority of them lacked co-stimulatory/inhibitory molecules, we used PB ILCs as a model to explore the antigen-presenting capacity of human HLA-DR+ ILCs.

To investigate whether PB ILCs are able to take up and process full-length protein, we adopted the DQ-Ovalbumin (DQ-OVA)

system. PB CD117⁻ ILCs, ILC2, HLA-DR⁻, and HLA-DR⁺ ILC3-like cells as well as Lin⁺HLA-DR^hi cells were FACS sorted and incubated for 4 h with DQ-OVA at 37 °C. All analyzed cell subsets were able to take up and process DQ-OVA, which was efficiently prevented by incubation of the cells on ice (Fig. 2c, d). As expected, the strongest uptake and degradation was observed in the Lin⁺HLA-DR^hi population, which contains professional antigen-presenting cells (APCs) such as B cells, monocytes/macrophages, and dendritic cells (DCs). PB CD117⁻ ILCs, ILC2, HLA-DR⁻, and HLA-DR⁺ ILC3-like cells displayed similar MFI values following the incubation with DQ-OVA (Fig. 2c, d). Taken together, PB ILCs are able to take up and degrade full protein antigens via proteolysis. Such degradation is equally efficient in all ILC subsets, independent of HLA-DR expression.

**PB ILCs fail to induce T-cell antigen recall responses**. In contrast to naive T helper cells, memory CD4⁺ T cells might not necessarily require co-stimulation[29,30]. We therefore examined the possibility that ILCs could induce memory CD4⁺ T-cell responses to the human cytomegalovirus (CMV) immunedominant protein pp65. CMV was selected due to its widespread prevalence in the human population in addition to high frequencies of CMV-specific memory CD4⁺ T cells in exposed individuals[31]. To enrich for the CMV-specific T cells, we used the method previously described by Bacher et al.[32]. Briefly, PBMCs from CMV⁺ donors were stimulated with CMV-pp65 peptide pool in the presence of CD40 blocking antibody, preventing the CD40-CD40L interactions and subsequent CD40L internalization by antigen-specific CD4⁺ memory T cells. Thereafter, CMV-pp65-reactive memory T cells were enriched using magnetic beads targeting CD40L-expressing cells. CD40L⁺ CD4⁺ T cells were expanded in IL-2 (Fig. 3a). Expanded memory T cells were confirmed to be specific for CMV-pp65 since they did not respond to an unrelated antigen (Supplementary Fig. 7).

To investigate their antigen-presenting properties, CMV-pp65 protein-loaded ILCs or CD3⁻ PBMCs were co-cultured with autologous CMV-pp65-specific CD4⁺ memory T cells at a 1:2 ratio (Fig. 3b, c). Protein-loaded CD3⁻ PBMCs, a significant proportion of which was constituted by professional CD40-expressing APCs (Fig. 3d), were able to evoke robust cytokine production and CD40L expression by T cells. However, ex vivo PB ILCs failed to induce T-cell recall responses against CMV-pp65 in 4 out of 5 analyzed donors (Fig. 3b, c). Thus, despite HLA-DR expression on a subset of CD117⁻ ILCs and ILC3-like cells from PB of healthy individuals, these cells were not able to efficiently elicit a CMV-pp65-specific memory T helper-cell response.

**Cytokines regulate APC characteristics of PB ILCs**. We previously described that the tonsil-derived HLA-DR⁺ ILC3 subset is enriched for cells expressing transcripts encoding the IL-1β receptor (IL1R1)[14]. IL-1β production is tightly regulated by inflammasome activation and is frequently paralleling the production of IL-18. Based on our previous observations and the hypothesis that inflammasome-associated cytokines might induce antigen-presenting properties in ILCs, we set out to investigate the effect of IL-1β and IL-18 on the expression of HLA-DR and co-stimulatory molecules on FACS sorted PB ILCs in a time-course experiment. In addition, we determined whether a combination of the ILC3-polarizing cytokines IL-23 and IL-1β would have any effect on HLA-DR and co-stimulatory molecule expression by PB ILCs. Highlighting the in vivo relevance of these experiments, transcripts of all of the above-mentioned cytokines were detected in non-affected as well as cancer-associated colon tissue (Supplementary Fig. 8).

After 24 h of cytokine treatment, no upregulation of HLA-DR and co-stimulatory molecules was observed on ILC2 and ILC3-like cells (Supplementary Fig. 9). The viability of CD117⁻ ILCs was strongly impaired by these culture conditions, which forced us to exclude these cells from further analysis.

After 3 days, and more pronounced after 5 days of culture, ILC3-like cells significantly upregulated HLA-DR, CD70, CD80, and CD86 in response to IL-1β (Fig. 4a, b). At both time points, IL-18 exposure led to similar, but less pronounced, differences and the combination of IL-1β plus IL-18 did not show any additive or synergistic effects as compared with IL-1β alone (Fig. 4a). The combination of IL-1β and IL-23 led to a significant reduction of HLA-DR, CD80 and CD70 expression on ILC3-like cells, compared with IL-1β treatment (Fig. 4a). Since IFN-γ is known to be a positive inducer of MHCII expression[33], we investigated its effect on HLA-DR and co-stimulatory molecule expression on PB ILC3-like cells. While we were able to detect a dose-dependent upregulation of HLA-DR, no CD70, CD80, or CD86 expression was observed following 5 days of IL-2 plus IFN-γ stimulation (Supplementary Fig. 10a, b). The level of HLA-DR and CD86 expression on PB ILC3-like cells following 5-day IL-1β treatment was comparable to that of ex vivo professional APCs (Fig. 3d, Supplementary Fig. 5).

Compared with PB ILC3-like cells, ILC2 showed a delayed expression of HLA-DR and co-stimulatory molecules, as these were not significantly upregulated in response to IL-1β or IL-18 until day 5 of culture (Fig. 4c, d). Unlike ILC3-like cells, PB ILC2 did not express HLA-DR or co-stimulatory molecules upon IFN-γ stimulation (Supplementary Fig. 10c).

We also examined a set of well-known co-stimulatory or co-inhibitory molecules on the surface of PB ILC2 or ILC3-like cells, none of which were expressed ex vivo or following 5 days IL-2 plus IL-1β exposure (Supplementary Fig. 11). In addition, IL-1β had no effect on naive or memory CD4⁺ T cells (Supplementary Fig. 12).

Phenotypically, PB ILC3-like cells exposed to IL-1β for 5 days either alone or in combination with IL-18 or IL-23 still expressed CD127 and CD161, albeit to a lesser extent (Supplementary Fig. 13a, b). With the exception of IL-2 mono-treatment, ILC2 maintained high expression of CD161, while exhibiting some downregulation of CD127 and CRTH2 (Supplementary Fig. 13B). Ruling out significant differentiation/plasticity toward NK cells, both ILC2 and ILC3-like cells maintained high expression, or even upregulated CD117 (Supplementary Fig. 13b) and lacked NKG2A or CD94 (Supplementary Fig. 13c). In line with previous observations that IL-1β is an important co-factor driving human ILC polarization toward different lineages[34–39], IL-1β-expanded ILC3-like cells maintained high expression levels of RORγT, intermediate expression of T-bet and upregulated GATA3 (Supplementary Fig. 13d). Similarly, ILC2 stimulated with IL-1β upregulated RORγT and T-bet expression, while displaying high levels of GATA3. Although IL-1β modified the master transcription factor expression of ILC2 and ILC3-like cells, they did not commit to a different ILC lineage, as additional cytokines such as IL-12 are necessary as switch factors to complete the terminal transdifferentiation[34–39].

As a known immunosuppressive cytokine, transforming growth factor β (TGF-β) not only plays a role in intestinal microbiota-immune cell crosstalk but has also been associated with negative prognosis in CRC patients with advanced tumors[40,41]. Moreover, we detected expression of TGFB1 and TGFB3 in all of the sub-anatomical colon regions analyzed (Supplementary Fig. 8). Therefore, we wanted to determine its influence on the antigen-presenting characteristics of PB ILCs. TGF-β counteracted the effect of IL-1β and significantly inhibited the upregulation of HLA-DR, CD80, and CD86 on ILC3-like cells (Fig. 4e), while T-cell inhibitory molecules PD-L1 and PD-L2

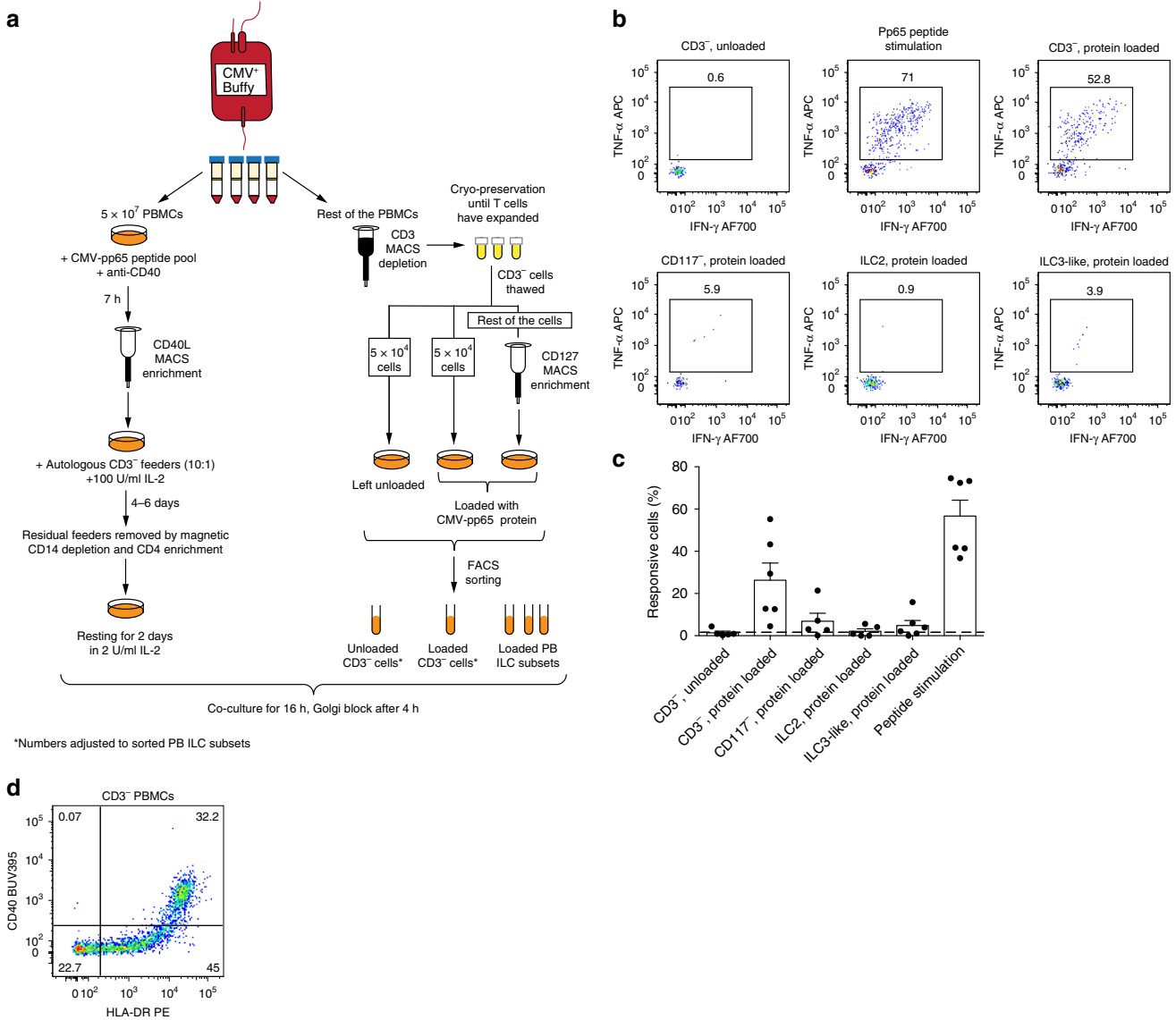

**Fig. 3 Antigen presentation by ex vivo PB ILCs. a** Experimental workflow to evaluate the antigen-presentation capacity of PB ILC subsets to autologous CMV-pp65-specific CD4[+] memory T cells. **b** Representative flow cytometric dot plots of IFN-γ and TNF-α production by expanded CMV-pp65-specific CD4[+] memory T cells after co-culture with autologous CMV-pp65 protein-loaded cell subsets, CD3[−] cells left unloaded, or in response to CMV-pp65 peptide stimulation. **c** Summarized data of frequencies of responsive (IFN-γ[+], TNF-α[+], and/or CD40L[+]) CMV-pp65-specific CD4[+] memory T cells after co-culture with autologous CMV-pp65 protein-loaded cell subsets or CMV-pp65 peptide stimulation. N (donors) = 6. Due to limited cell numbers, we could not generate data for all conditions for every donor; source data are provided as a source data file. Bars and error bars indicate mean and SEM. **d** Representative plot of HLA-DR and CD40 expression on CD3[−] cells present in the autologous CMV-pp65-specific co-culture system.

remained unchanged (Supplementary Fig. 14a). Interestingly, inhibitory effects on ILC2 were only detected for CD80 expression (Supplementary Fig. 14b), possibly due to differential cytokine receptor expression on the two cell types.

Taken together, IL-1β and to a lesser extent IL-18 enhance HLA-DR and co-stimulatory molecule expression on PB ILC3-like cells and ILC2. In our in vitro system, ILC3-like cells displayed a surface profile reminiscent of antigen presentation more rapidly and at higher frequencies compared with ILC2. We also observed an inhibitory effect of TGF-β on the antigen-presenting phenotype of PB ILC3-like cells, which was not as pronounced for ILC2.

**IL-1β/IL-18 induced APC characteristics are NF-κB-dependent.** We next examined the molecular pathways leading to HLA-DR

upregulation in PB ILC2 and ILC3-like cells following IL-1β or IL-18 stimulation. MHCII expression is tightly regulated by the class II transactivator (CIITA). CIITA transcription can be induced in various cell types of hematopoietic and non-hematopoietic origin through an IFN-γ-associated signaling pathway, involving subsequent STAT1 phosphorylation, dimerization and nuclear translocation[33]. Phosphorylation of STAT1 can occur at two molecular sites—Tyr701, which is indispensable for its homodimerization and DNA binding, and Ser727, which lies within the MAPK consensus motif and regulates transcriptional activity[42]. Interestingly, in human glioblastoma and fibrosarcoma cell lines, IL-1β was shown to induce phosphorylation of STAT1$_{Ser727}$ but not STAT1$_{Tyr701}$, which had a synergistic effect on IFN-γ-induced gene expression[43]. Moreover, Lee et al. have previously described an additional, NF-κB-dependent mechanism of HLA-DR expression in B cells independent of CIITA[44].

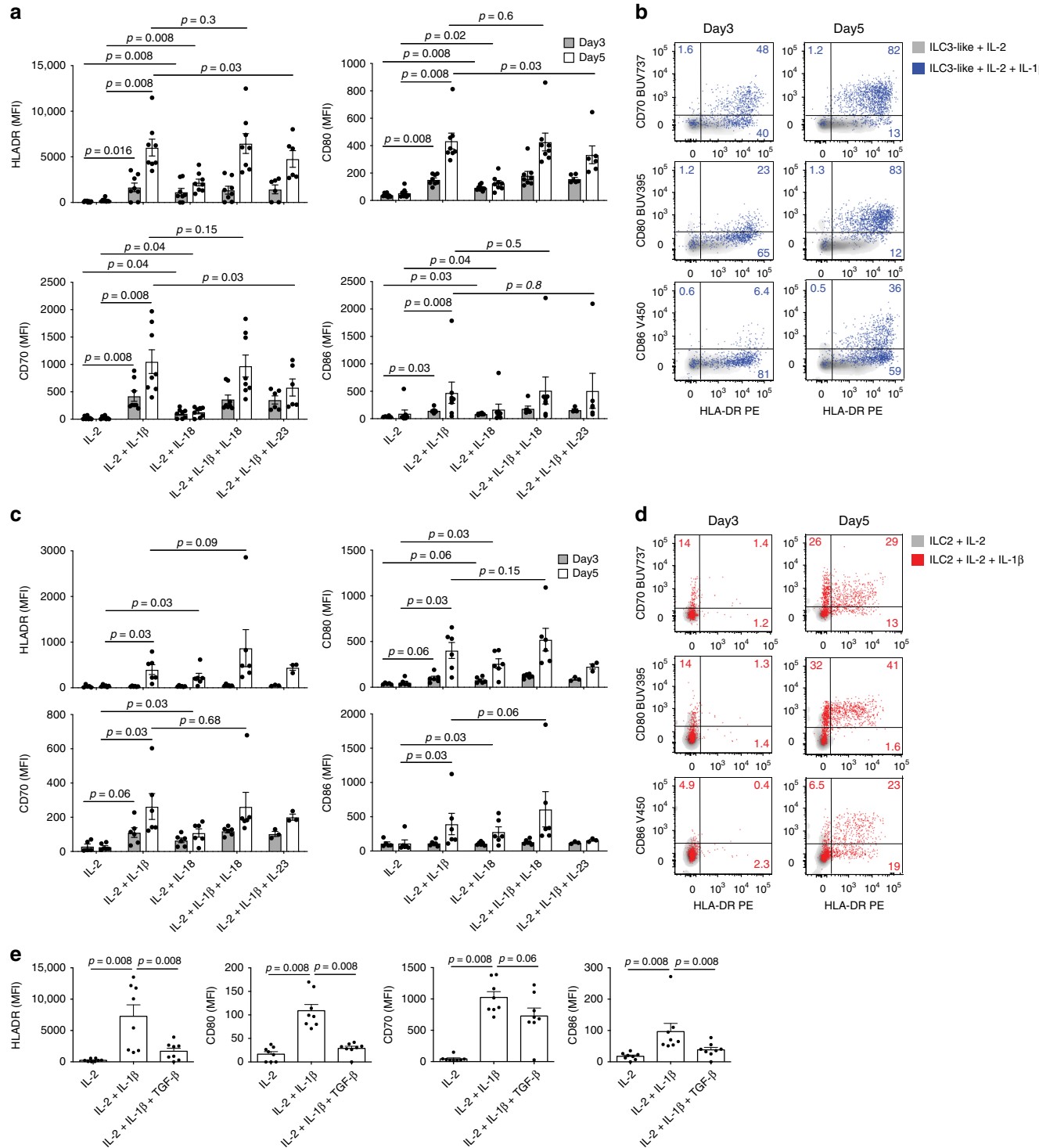

**Fig. 4 HLA-DR and co-stimulatory molecule expression on PB ILCs in response to cytokine treatment. a**, **c** Summarized data and **b**, **d** representative flow cytometric dot plots of HLA-DR, CD86, CD80, and CD70 expression by sort-purified ILC3-like cells (**a**, **b**) and ILC2 (**c**, **d**) after 3 and 5 days of treatment with the indicated cytokine combinations. N (donors) = 8 (for ILC3-like cells) and 6 (for ILC2). Due to limited cell numbers, we could not generate data for all conditions for every donor; source data are provided as a source data file. **e** HLA-DR, CD86, CD80, and CD70 expression by sort-purified ILC3-like cells after 5 days of IL-2, IL-2 plus IL-1β, or IL-2 plus IL-1β, and TGF-β treatment. N (donors) = 8; bars and error bars indicate mean and SEM; statistical significance was assessed using two-sided Wilcoxon matched-pairs signed rank test. Source data are provided as a source data file.

Therefore, we analyzed whether IL-1β or IL-18 treatment is able to induce STAT1$_{Ser727}$ or STAT1$_{Tyr701}$ phosphorylation and NF-κB activation in human PB ILC2 and ILC3-like cells (Fig. 5a, b). To avoid batch-related differences, stimulated cells were barcoded, pooled and further processed together (Supplementary

Fig. 15a). Stimulation of ILC2 and ILC3-like cells with IL-1β and IL-18 led to STAT1$_{Ser727}$ phosphorylation, which reached its maximum at 30–60 min of stimulation. As expected, no phosphorylation of STAT1$_{Tyr701}$ occurred, ruling out a role for STAT1 dimerization in the downstream effects of IL-1β. Furthermore,

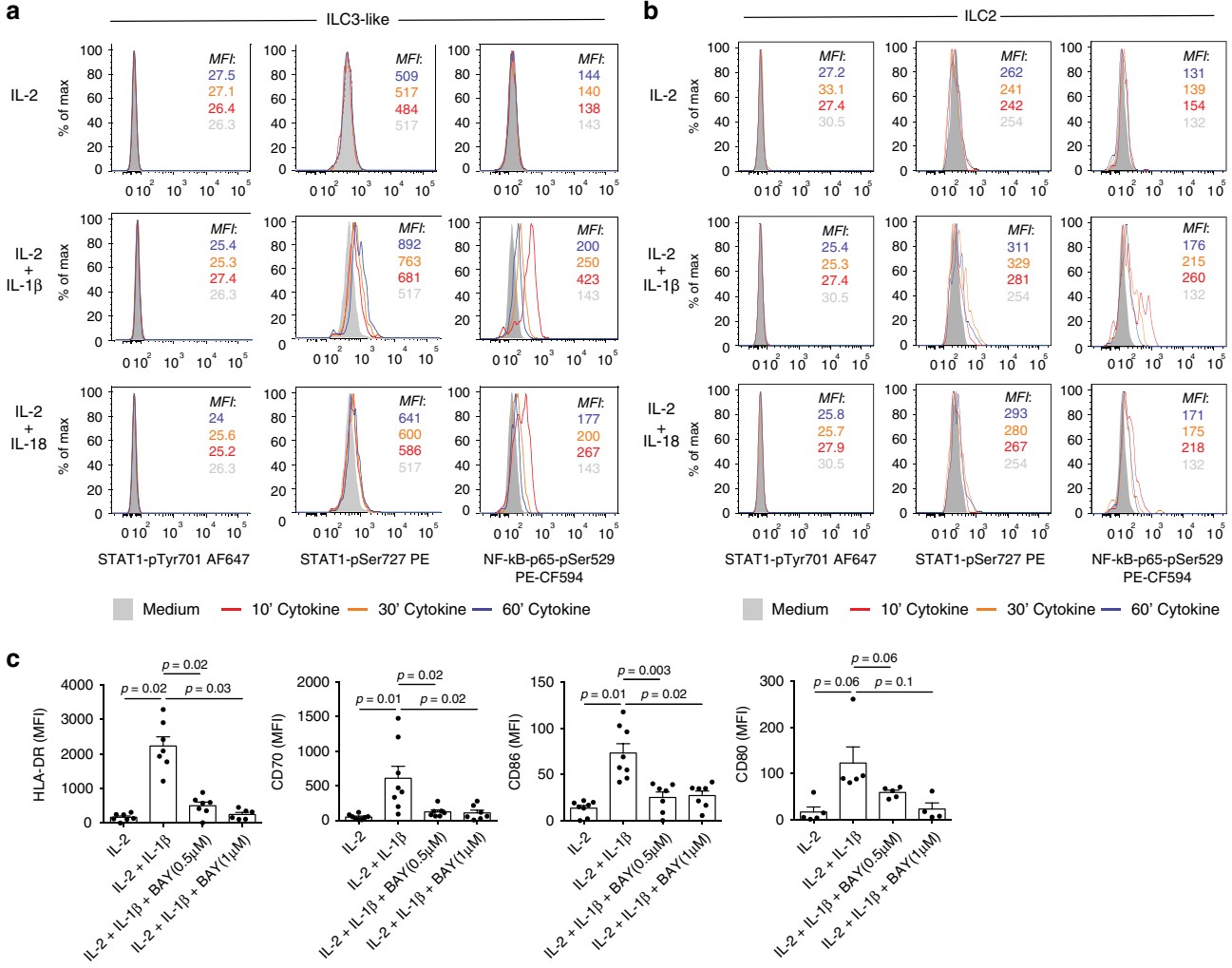

**Fig. 5 Mechanism driving HLA-DR and co-stimulatory molecule upregulation on PB ILCs after IL-1β or IL-18 treatment. a, b** Flow cytometric analysis of STAT1$_{Tyr701}$, STAT1$_{Ser727}$, and NF-κB-p65$_{Ser529}$ phosphorylation in sorted PB ILC3-like cells (**a**) and ILC2 (**b**) following 10, 30, or 60 min incubation with IL-2, IL-2 plus IL-1β, or IL-2 plus IL-18. Representative example of five independent experiments is displayed. **c** MFI of HLA-DR, CD70, CD80, and CD86 expression on sorted PB ILC3-like cells, following 72 h incubation with IL-2, IL-2 plus IL-1β, or IL-2 plus IL-1β in the presence BAY11-7082. *N* (donors) = 7 for HLA-DR, 8 for CD70 and CD86 and 5 for CD80; bars and error bars indicate mean and SEM; statistical significance was assessed using two-sided Wilcoxon matched-pairs signed rank test. Source data are provided as a source data file.

each of the two cytokines induced NF-κB activation in ILC2 and ILC3-like cells with a peak activity at 10 min (Fig. 5a, b; Supplementary Fig. 15b). In case of both STAT1$_{Ser727}$ phosphorylation and NF-κB activation, IL-1β was the more potent stimulus than IL-18 at these concentrations.

To investigate whether IL-1β-dependent HLA-DR and co-stimulatory molecule upregulation on PB ILCs is dependent on NF-κB signaling, we induced antigen-presenting phenotype on PB ILCs in the presence of NF-κB inhibitors—BAY11-7082 and BMS-345541[45–47]. For this, sort-purified PB ILC3-like cells were incubated for 3 days with IL-2 or IL-2 plus IL-1β in the presence or absence of BAY11-7082 or BMS-345541. Due to cytotoxicity of the inhibitors, cytokine treatment could not be prolonged beyond 3 days, which is before ILC2 acquire APC-properties (Fig. 4). Thus, the inhibition experiment could only be performed on PB ILC3-like cells. Inhibition of NF-κB signaling led to significantly reduced HLA-DR expression, and correlated inversely with the dose of the inhibitor (Fig. 5c, Supplementary Fig. 16). CD70 and CD86 expression was also significantly decreased after NF-κB inhibition with BAY11-7082 (Fig. 5c), and BMS-345541 inhibited the expression of CD70 and CD80 (Supplementary Fig. 16). In

conclusion, IL-1β and/or IL-18 stimulation of PB ILC3-like cells leads to NF-κB signaling, which, at least in the case of IL-1β, is needed for optimal HLA-DR and co-stimulatory molecule expression.

**IL-1β/IL-18-activated PB ILCs induce T-cell recall responses.**
We next investigated whether IL-1β/IL-18-induced upregulation of HLA-DR and co-stimulatory molecules on PB ILC2 and ILC3-like cells rendered these cells functional as antigen-presenting cells. For this purpose, ILCs were pre-activated with IL-1β/IL-18 in the CMV-pp65 antigen-presentation co-culture system described above (Fig. 6a). Expanded ILC subsets and control CD3$^-$ PBMCs were loaded with CMV-pp65 protein, whereas a fraction of CD3$^-$ PBMCs was left unloaded as control. Residual protein was removed by magnetic enrichment of loaded cells using CD45 microbeads. Thereafter, protein-loaded cells were co-cultured with CMV-pp65-specific memory T helper cells and antigen-specific T-cell responses were assessed (Fig. 6a-d). Activated human T cells upregulate surface HLA-DR expression and have been shown to facilitate antigen presentation[48].

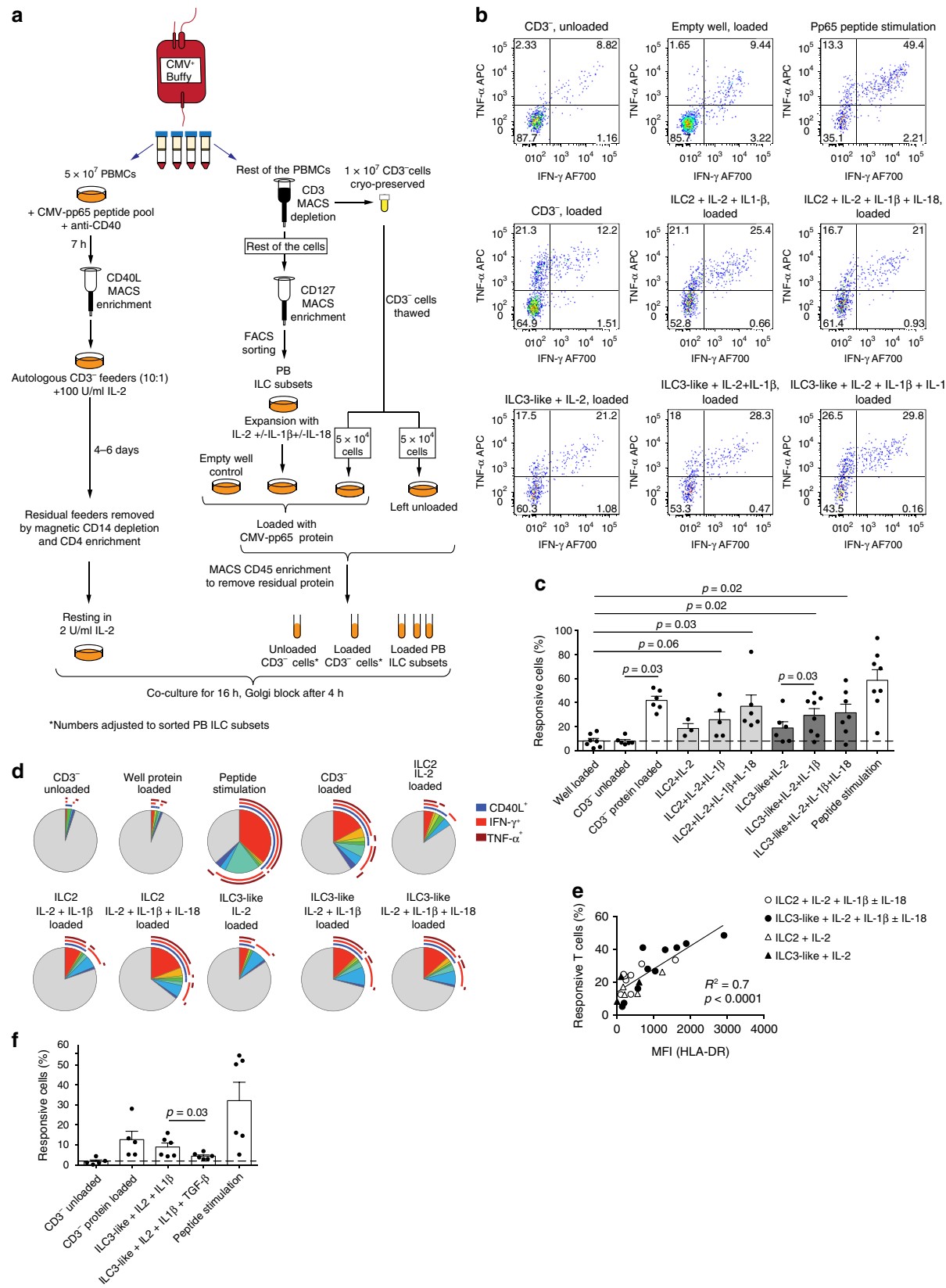

Therefore, to exclude the possibility that expanded CMV-pp65-specific memory T cells took up and presented antigen without APC assistance, an empty well was loaded with protein and treated as control, identically to the loaded cells (Fig. 6b–d). Aligning with the upregulation of HLA-DR and co-stimulatory

molecules, ILC2 and ILC3-like cells cultured with IL-2 plus IL-1β with or without IL-18 were able to efficiently induce antigen-specific responses by CMV-pp65-specific T helper cells (Fig. 6b–d). Multi-functionality of the responding CMV-pp65-specific memory T helper cells was analyzed by Boolean gating

**Fig. 6 Antigen presentation by PB ILCs following cytokine treatment. a** Experimental workflow to evaluate the antigen-presentation capacity of expanded PB ILC2 and ILC3-like cells. **b** Representative plots of cytokine production by CMV-pp65-specific CD4$^+$ memory T cells after co-culture with autologous CMV-pp65 protein-loaded cell subsets, CD3$^-$ cells left unloaded, or in response to CMV-pp65 peptide stimulation. To exclude antigen presentation of residual protein by T cells, an empty well was loaded with CMV-pp65-protein and treated further in the same way as expanded cells. **c** Summarized data of frequencies of responsive (IFN-γ$^+$, TNF-α$^+$, and/or CD40L$^+$) CMV-pp65-specific CD4$^+$ memory T cells after co-culture with indicated autologous cell subsets. N (donors) = 8. Due to limited cell numbers, we could not generate data for all conditions for every donor; source data are provided as a source data file. Bars and error bars indicate mean and SEM; statistical significance was assessed using two-sided Wilcoxon matched-pairs signed rank test. **d** Mean frequency of responsive CMV-pp65-specific CD4$^+$ memory T cells positive for combinations of IFN-γ$^+$, TNF-α$^+$, and CD40L$^+$ (indicated by arches) after co-culture with indicated autologous cell subsets. Size of each slice indicates frequency of cells positive for the given cytokine combination. N (donors) = 7. **e** Correlation between MFI of HLA-DR expression on expanded PB ILC subsets and percent of responsive T cells in autologous CMV-pp65-specific co-cultures (Pearson's correlation analysis, 95% confidence interval, two-tailed). **f** Summarized data of frequencies of responsive (IFN-γ$^+$, TNF-α$^+$, and/or CD40L$^+$) CMV-pp65-specific CD4$^+$ memory T cells after co-culture with indicated autologous sort-purified cell subsets. N (donors) = 6 for ILC subsets and 5 for CD3$^-$ cells; bars and error bars indicate mean and SEM; statistical significance was assessed using two-sided Wilcoxon matched-pairs signed rank test. Source data are provided as a source data file.

and SPICE software[49]. Pertaining to the cytokine composition, no significant differences were detected in conjunction with any of the antigen-presenting cell subsets (Fig. 6d). CD3$^-$ PBMCs and all expanded ILC subsets induced expression of CD40L as well as IFN-γ and TNF-α production by CMV-pp65-specific memory CD4$^+$ T cells, with IFN-γ being the most predominant cytokine (Fig. 6d).

Importantly, in most of the donors tested, 5-day incubation with IL-2 alone enabled PB ILC2 and ILC3-like cells to induce antigen-specific responses by memory T cells (Fig. 6c, d). In line with this, PB ILC2 and ILC3-like cells upregulated HLA-DR expression following 5-day stimulation with IL-2 alone (Supplementary Fig. 17a). Under these conditions, however, the cells did not express any co-stimulatory molecules (Fig. 4), underlining that co-stimulation may not be essential to induce cytokine response by antigen-specific memory T cells as reported by others[29,30]. Moreover, the magnitude of the observed T-cell response correlated with the HLA-DR expression levels on the antigen-presenting ILC populations (Fig. 6e). Of note, although CD3$^-$ cells served as a control in this experiment, the antigen-presenting capacity of these frozen-and-thawed cells was likely negatively impacted hampering a direct comparison with expanded ILCs.

To exclude that observed T-cell responses were the result of contamination by B cells or DCs in our ILC cultures, we assessed CD40 expression on CD3$^-$ cells in the co-cultures. While CD40 was readily detected on CD3$^-$ PBMCs, cytokine-expanded ILC subsets did not display any CD40 expression (Fig. 3d, Supplementary Fig. 17b). Thus, the antigen-presentation activity observed in our assays can be attributed to ILCs and was not a consequence of B-cell or DC-mediated confounding effects.

Next we investigated whether TGF-β-mediated inhibition of antigen-presenting properties of ILC3-like cells translated into functionally impaired capacity to induce T-cell response. Indeed, treatment with TGF-β resulted in significantly diminished T-cell responses (Fig. 6f). This effect is likely a consequence of reduced expression of HLA-DR and co-stimulatory molecules and not of modified antigen uptake and processing, as IL-2 ± IL-1β ± TGF-β expanded ILC3-like cells did not display significant differences in proteolytic digestion of DQ-OVA protein (Supplementary Fig. 18b).

Taken together, exposure to IL-1β induces effective antigen-presentation properties in PB ILC2 and ILC3-like cells, enabling them to prompt cytokine responses by CMV-pp65-specific memory T helper cells. In PB ILC3-like cells this process is inhibited by TGF-β.

**IL-1β/IL-18 induce APC features of intestinal ILCs.** We next investigated whether the effect of inflammasome-associated cytokines is restricted to circulating ILCs or whether intestinal

ILCs also have the intrinsic propensity to develop APC-like features upon IL-1β plus IL-18 exposure. Purified ILCs from non-affected colon, tumor border and central tumor tissue were cultured in the presence of IL-2 plus IL-1β and IL-18 for 7−9 days and the resulting expression of HLA-DR and T-cell co-stimulatory molecules was assessed (Fig. 7). Although no significant differences were detected in connection with the sub-anatomical origin of ILCs, inflammasome-associated cytokines induced HLA-DR, CD70, CD80, and CD86 upregulation on ILCs from all donors analyzed (Fig. 7). These findings underline that IL-1β- and IL-18-based induction of APC-like features of ILCs is not restricted to circulating immature PB ILC3-like population, but can also be observed in tissue-resident intestinal ILCs (Fig. 8).

**Discussion**
Although MHC-dependent ILC-T-cell crosstalk has been demonstrated in several mouse models[18–22], the antigen-presenting properties of human ILCs remain underexplored. In the present study, we show that human colorectal tumors contain ILCs with increased HLA-DR expression, albeit lacking T-cell co-stimulatory molecules in most donors. HLA-DR$^+$CD127$^+$ ILCs co-localize with T cells in non-affected colon tissue as well as at the tumor border, suggesting the possibility of physical interaction between the two cell types. ILC-based suppression of T-cell responses to intestinal commensal bacteria has been previously demonstrated in mesenteric lymph nodes in mice[20,21]. Since we did not detect any co-stimulatory molecules on the surface of ILCs from the healthy colon and in only three cases of tumor-associated tissue, an analogous mechanism may be in place in humans. Indeed, in a back-to-back publication[50], Lehmann et al. demonstrate that in contrast to the spleen, where ILC3 are efficient T-cell activators, the gut microenvironment suppresses the antigen-presenting capacity of ILC3. However, due to differences in T-cell composition and the overall intestinal microenvironment in humans and transgenic mice, activation of tissue-resident memory CD4$^+$ T cells by antigen-presenting ILCs in the human mucosa cannot be ruled out, especially in the context of inflammation or cancer. In support of this, it was recently shown that during acute colitis, TL1A-activated ILCs prompt antigen-specific T-cell responses in the intestinal mucosa[23]. In line with this study, we showed that intestinal ILCs have an intrinsic capacity to acquire antigen-presentation characteristics, depending on the cytokine microenvironment. This observation was supported by the three cases of colorectal tumors where we identified low levels of CD86.

We used PB ILCs as a model to unravel how the antigen-presenting capacity of human ILCs is regulated by CRC-associated cytokines, i.e., IL-1β, IL-18, IFN-γ, and TGF-β[51,52]. Transcripts of these cytokines were confirmed to be present in

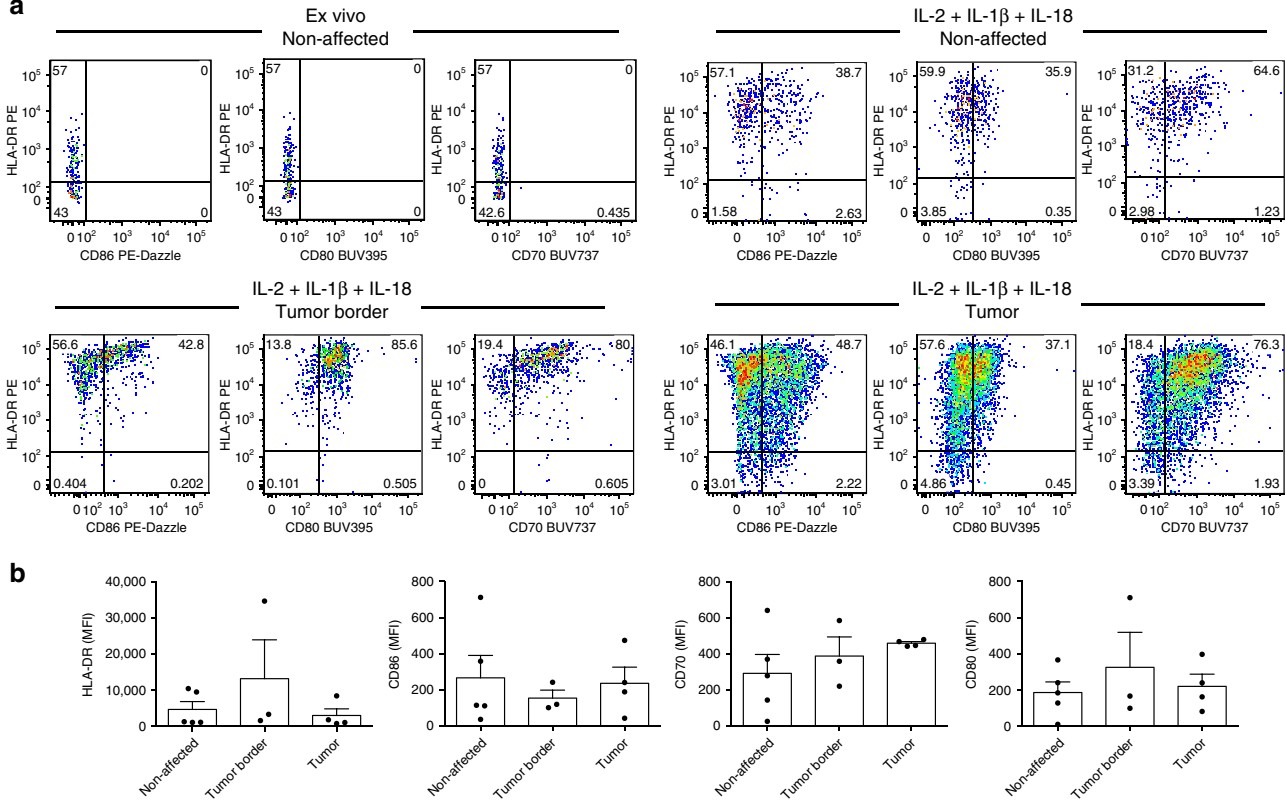

**Fig. 7 HLA-DR and co-stimulatory molecule expression on intestinal ILCs ex vivo or following IL-2 plus IL-1β and IL-18 stimulation. a** Representative plots and **b** MFI of HLA-DR, CD70, CD80, and CD86 expression on ex vivo (**a**) and sort-purified (**a**, **b**) ILCs isolated from non-affected colon, tumor border and central tumor areas expanded for 7–9 days in the presence of IL-2 plus IL-1β and IL-18. N (patients) = 5. Not all sub-anatomical regions could be obtained for every donor; source data are provided as a source data file. Bars and error bars indicate mean and SEM.

non-affected and tumorous intestinal tissue. Although we did not detect significant differences in the transcription of the corresponding cytokine genes, other larger studies have reported increased expression of *IL1B*[53,54], *IL23A*[54], and *TGFB1*[55] in colorectal tumors. We acknowledge that the cytokine concentrations used in our in vitro assays may not necessarily reflect the actual amount present in situ, which are hard to determine.

IL-1β-stimulated PB ILC2 and ILC3-like cells upregulated HLA-DR and co-stimulatory molecules and induced antigen-specific response in CMV-pp65-specific memory helper T cells. This observation is in keeping with the results reported by von Burg et al., where IL-1β-driven upregulation of MHCII and co-stimulatory molecules on mouse splenic ILC3 enabled them to induce the proliferation of naive T cells[19]. Conversely, Hepworth et al. showed that TLR-agonists, IL-23, IL-1β, and IFN-γ did not upregulate MHCII or co-stimulatory molecules on ILCs derived from mLN or colonic mucosa in mice[20], which contrasts with the observations we made for IL-1β and IL-18 using human tissue. Furthermore, as opposed to mouse ILC3[19], human IL-1β-expanded PB ILC2 or ILC3-like cells did not express CD40. Together, this suggests that the mechanisms regulating the antigen-presenting capacity of ILCs are not conserved across these two species, emphasizing the need for studies of human tissues. Of note, although IL-1β cultured cells started to express transcription factors associated with all non-NK lineages (RORγT, GATA3, and T-bet), as previously reported[37], full trans-lineage differentiation requires polarizing cytokines such as IL-12 (ILC1)[36] and IL-23 plus TGF-β (ILC3)[35,38,39].

While IL-1β potently induced APC-like features in both ILC2 and ILC3-like cells, IL-18 addition prompted no cumulative

effects. Both cytokines promoted NF-κB activation in PB ILC2 and ILC3-like cells. Lee et al. have previously described an NF-κB-dependent mechanism of HLA-DR expression using a human B-cell line, where NF-κB was shown to bind to a conserved motif upstream of the S-X-Y module and drive HLA-DR expression independent of CIITA[44]. As NF-κB inhibition almost completely abrogated HLA-DR expression on stimulated PB ILC3-like cells, IL-1β stimulation of human PB ILCs might trigger a similar mechanism such as that described for B cells by Lee and colleagues.

Although IL-1β levels gradually increase with the transformation of healthy colon tissue to adenoma and further to CRC[56], tumor-associated ILCs displayed CD86 expression only in three out of thirteen CRC cases analyzed. This was not linked to an intrinsic inability of intestinal ILCs to provide T-cell help, as in vitro IL-1β- and IL-18-stimulated ILCs from non-affected as well as cancer tissue readily upregulated HLA-DR and co-stimulatory molecules. The lack of CD70, CD80, or CD86 expression on ILCs in colorectal tumors could, therefore, be the consequence of signals counteracting the effect of inflammasome-associated cytokines in the local microenvironment. Indeed, TGF-β efficiently dampened IL-1β-induced HLA-DR and co-stimulatory molecule upregulation on PB ILC3-like cells in vitro, representing one possible counter-regulatory mechanism. Also, IL-23, one of the key cytokines driving ILC3 function and differentiation[10,57], exerted a suppressive effect on HLA-DR and co-stimulatory molecule expression on IL-1β-stimulated PB ILC3-like cells. IL-1β, IL-18, and IL-23 are produced by activated DCs and macrophages in response to microbial components and other danger signals[58,59]. However, there are differences in

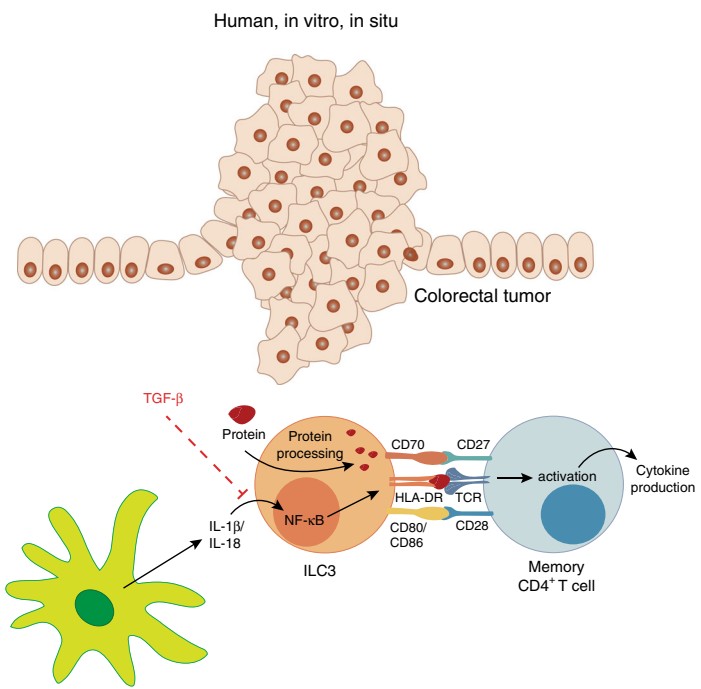

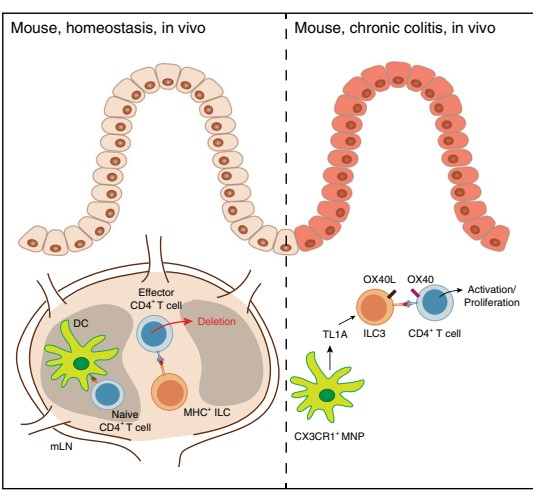

**Fig. 8 Proposed model: the role of ILC3 in regulation of intestinal CD4$^+$ T-cell responses.** While it is known that naive CD4$^+$ T cells are primed in mesenteric lymph nodes (mLN), several studies demonstrated ILC3-based regulation of effector CD4$^+$ T-cell responses in mice. MHCII$^+$ ILC3 were shown to accumulate between T- and B-cell zones in mouse mLNs, co-localizing with emigrating commensal bacteria-specific effector CD4$^+$ T cells and perpetrating the depletion of the latter[20, 21]. In the setting of experimental chronic colitis, TNF-like ligand 1A (TL1A) produced by mononuclear phagocytes (MNPs) was observed to induce OX40L expression on ILC3, which in turn promoted T-cell activation and immunopathology[23]. In the present study we show that ILCs in human colorectal tumors display increased HLA-DR expression and co-localize with T cells in situ. Further in vitro analysis revealed that the inflammasome-associated cytokines IL-1β and IL-18 could drive the expression of HLA-DR and co-stimulatory molecules on PB ILCs in an NF-κB-dependent manner, while TGF-β potently inhibited these antigen-presenting properties. IL-1β- and IL-18-activated ILCs were able to induce memory CD4$^+$ T-cell responses in an autologous co-culture system following the uptake of full protein antigen. Hence, we would like to propose that cytokines present in the human colorectal tumor microenvironment might be able to modulate the antigen-presenting properties of intestinal ILCs with potential consequences for anti-tumor immune responses.

stimulatory mechanisms for inflammasome activation and IL-23 production. For instance, while CD40-CD40L signaling is a negative regulator of inflammasome activation[58], it is also known to induce IL-23 production[59]. Furthermore, TGF-β can be produced by several cell types and converted to its bioactive form by specific DC subsets in the human intestine[41,60,61]. We observed CD127$^+$ ILCs co-localizing with CD45$^+$HLA-DR$^{hi}$ cells—which are likely to represent mononuclear phagocytes (MNPs)—as well as T cells around the crypts in our immunohistological analysis. Such a pattern of cellular localization would enable DCs to rapidly fine-tune the antigen-presenting functions of CD127$^+$ ILCs. Whether this cell-to-cell interaction would directly influence antigen-specific memory T-cell responses and the effects thereof remains to be elucidated. A similar concept of an MNP-ILC-T-cell regulatory axis has been recently proposed by Castellanos et al.[23], where MNP-derived TL1A induced OX40L expression on ILC3, which in turn promoted T-cell activation and colitis pathogenesis in mice.

Nonetheless, it remains unclear whether co-stimulation is needed for activation of memory CD4$^+$ T cells[29,30]. We observed an increased antigen-presentation capacity of PB ILC2 and ILC3-like cells cultured with IL-2 alone for 5 days. This was paralleled by an increase in HLA-DR expression, although in the absence of co-stimulatory molecules. Contrastingly, Oliphant et al. described that human ILC2 express HLA-DR, CD80, and CD86 following expansion with 100 U/ml IL-2 and gamma-irradiated PBMCs[18]. Differences observed could be due to the presence of IL-1β-producing feeders in the afore-mentioned system[34]. Since

intestinal ILCs display higher HLA-DR expression than their PB counterparts, it is possible that the former may activate local memory CD4$^+$ T-cell subsets.

Our findings extend existing knowledge concerning the antigen-presenting properties of ILCs, with an emphasis on human memory T-cell responses. A deeper understanding of ILC-T-cell interactions and how they are influenced by the immediate tissue microenvironment could be instrumental for designing future immunotherapeutic approaches. This applies not only to cancer but also other pathologies where inflammation might unleash the intrinsic capacity of ILCs to act as promoters of CD4$^+$ T-cell responses.

## Methods

**Study design and patient samples.** Peripheral blood mononuclear cells (PBMCs) were isolated from buffy coats, or from whole venous blood of healthy donors. Blood collection was approved by the Swedish Ethical Review Authority. Paired peripheral blood and non-affected as well as cancerous gut tissue was obtained from patients undergoing colorectal cancer surgery. Patient information can be found in Supplementary Table 1. Written informed consents were obtained from the patients and the sample collection was approved by the Swedish Ethical Review Authority.

**Multicolor immunofluorescence microscopy.** Intestinal resection material was embedded in OCT compound (HistoLab), frozen on dry ice and stored at −80 °C. Five-millimeter-thick cut frozen sections were thawed and fixed in ice cold acetone for 5 min followed by one wash with Tris Buffer Saline (TBS) for 10 min. Fixed sections were consequently analyzed for the presence of isolated lymphoid follicles under the light microscope (Supplementary Figs. 3, 4). Sections were incubated with Image-iT® FX signal enhancer (Life Technologies) for 30 min at room

temperature, washed once in TBS and blocked with Background Buster (Innovex Biosciences) for 15 min at room temperature. Endogenous biotin was then blocked using Abcam Endogenous Avidin/Biotin Blocking Kit (Abcam).

A stepwise protocol was used to avoid cross reactivity of mouse and rat secondary antibodies. Following blocking steps, sections were incubated with the primary antibodies targeting CD127 (clone eBioRDR5, eBioscience) and HLA-DR (clone L243, Biolegend) for 1 h at room temperature. The primary antibodies were detected with the corresponding goat anti-mouse IgG1 AF488 and goat anti-mouse IgG2a AF647 (both Invitrogen) for 30 min at room temperature.

Sections were then incubated with primary antibodies targeting CD3e (rabbit, clone EP449E, Abcam) and CD45 (rat, clone YAML501.4, Invitrogen) overnight at 4 °C and detected with the corresponding secondary goat anti-rat AF680 and goat anti-rabbit AF555 (both Invitrogen) for 30 min at room temperature. Sections were incubated with DAPI at 0.0002% w/v together with secondary antibodies. Following staining, background autofluorescence was quenched using TrueBlack® Lipofuscin Autofluorescence Quencher (Biotium) for 5 min followed by two brief washes, then washed once for 10 min in TBS. The slides were mounted using Prolong Diamond (Invitrogen).

Spectral images were acquired with a Nikon confocal microscopy system (Nikon A1R instrument; Nikon) equipped with a 32-channel spectral detector using a ×20 Nikon air objective. Reference spectra from gut sections stained with single fluorophores for AF488, AF555, AF647, AF680 as well as DAPI were individually acquired and applied to extract and produce single-color grayscale images for each fluorophore from the spectral images using the linear unmixing function in Nikon Elements software (Nikon). Isotype controls for each primary antibody and fluorescence minus one control for each secondary antibody were used to exclude non-specific primary antibody binding and assess any fluorophore spill over. Images were generated using Cytosketch (CytoCode).

**Quantification of microscopy findings.** Quantification of microscopy findings was undertaken using Imaris (Oxford Instruments). Cells were identified using the Spots Detection Function and manually confirmed for accuracy. ILCs were identified as CD127+CD45+CD3− and based on cellular morphology; T cells were identified as CD3+CD45+ and based on cellular morphology. Crypts and cellular aggregates were manually drawn using the "Surfaces" function in Imaris, and subsequently distances were determined using the Distance Tranformation Xtension. Cell-to-cell proximity was determined using the Spots Colocalize Xtension in Imaris. Statistical calculations were performed using Imaris.

**Cell isolation and sorting.** PBMCs were isolated using Ficoll gradient density centrifugation. To isolate intestinal mononuclear cells (MNCs) muscle and adipose tissues were removed and remaining gut tissue was mechanically disrupted, followed by enzymatic digestion with 250 μg/ml DNase and collagenase II at 37 °C and magnetic stirring at 450 rpm for 45 min. Cell suspension was filtered using 70 μm cell strainer. Mononuclear cells were isolated using Ficoll gradient density centrifugation.

In case of subsequent FACS purification of ILCs, MNCs were depleted of T cells using anti-CD3 microbeads and LD columns and enriched using CD127 microbead Kit and LS columns (all Miltenyi Biotech) according to manufacturer's instructions. CD3−CD127+ MNCs were surface stained for 30 min at room temperature. ILCs were sort-purified as lineage−CD45+CD3−CD127+CD161+ lymphocytes, with ILC1 being CD117−CRTH2−, ILC2—CRTH2+, and ILC3—CD117+CRTH2− cells. A full list of antibodies used can be found in Supplementary Table 2. Cells were sorted using BD FACSAria™ Fusion Cell Sorter or SONY MA900 Multi-Application Cell Sorter.

**RNA isolation and microarray analysis of gene expression.** Muscle and adipose tissues were removed and a small piece (~2 × 2 mm) of the remaining non-affected as well as cancer-associated colon tissue was preserved in RNAlater (Ambion) at 4 °C for a maximum of 2 weeks. Subsequently, RNAlater buffer was removed and total RNA isolated using RNeasy Mini Kit (Qiagen). Microarray analysis was performed using the human Clariom™ D Assay, applying the standard Affymetrix platform. The ".cel" files were preprocessed using Bioconductor packages in R such as Oligo[62] and affycorectools[63]. For transcript annotation clariomdhumantranscriptcluster.db[64] library from Bioconductor package in R was used. Using oligo package, we imported the ".cel" files in R. Subsequently, we carried out robust multichip averaging preprocessing strategy (oligo-rma)[65] for background subtraction, quantile normalization and summarization of the expression arrays.

**In vitro cell culture of ILCs and pathway inhibition.** Sorted PB or intestinal ILCs were cultured in Yssel's-supplemented IMDM with 1% normal human serum and 10 U/ml IL-2 (PeproTech). In case of stimulation, cells were additionally treated with 50 ng/ml IL-1β (R&D Systems), 50 ng/ml IL-23 (R&D Systems), 50 ng/ml IL-18 (R&D Systems), 20 ng/ml TGF-β1 (R&D Systems), or IFN-γ (PeproTech) for 3−9 days.

To inhibit NF-κB signaling pathway BAY11-7082 (Santa Cruz Biotechnology) and BMS-345541 (Sigma-Aldrich) were used. Sorted ILCs were stimulated for 3 days with 10 U/ml IL-2 plus 50 ng/ml IL-1β in the presence or absence of the corresponding inhibitor. Different concentrations of the inhibitors were used:

0.5 μM and 1 μM of BAY11-7082, and 1, 2, and 4 μM of BMS-345541. Subsequently the cells were harvested and HLA-DR and co-stimulatory molecule expression were accessed by flow cytometry. Nonviable cells were excluded from the analysis based on the dead cell marker and CD45 staining.

**Flow cytometry.** Cell surface was stained at room temperature for 30 min. In case of subsequent intracellular cytokine staining, cells were fixed for 10 min with 2% paraformaldehyde and permeabilized using 1x Permeabilizing Solution 2 (BD Biosciences) at room temperature for 10 min. Cytokine staining was performed at room temperature for 30 min. Transcription factors were stained using eBioscience™ FoxP3/Transcription Factor Staining Buffer Set (Thermo Fisher Scientific) according to manufacturer's instructions. A full list of antibodies used can be found in Supplementary Table 2. Stained samples were acquired on BD LSR Fortessa™ equipped with 355-, 405-, 488-, 561-, and 639-nm lasers. Analysis was performed using FlowJo v. 9.9 (TreeStar).

**DQ-OVA uptake.** Sort-purified ILC1, ILC2, and ILC3 as well as Lineage+HLA-DRhi cells isolated from buffy coats were cultured at 37 °C in Yssel's-supplemented IMDM plus 1% normal human serum with or without 10 μg/ml DQ-OVA (Invitrogen). As a negative control cells were cultured in the presence of DQ-OVA on ice. After 4 h cells were harvested and further analyzed using flow cytometry.

**Generation and re-stimulation of T-cell lines.** CMV-pp65-specific memory T cells were isolated and expanded as previously described by Bacher et al.[32] with minor modifications. $5 \times 10^7$ PBMCs from CMV+ buffy coats were stimulated in Yssel's-supplemented IMDM with 1% normal human serum with PepTivator CMV-pp65 in the presence of anti-CD40 (both Miltenyi) for 7 h. Thereafter cells were harvested and CMV-pp65-reactive cells were enriched using human CD154 MicroBead Kit (Miltenyi). CMV-pp65-specific T-cell lines were expanded for 4–6 days in the presence of CD3− gamma-irradiated autologous feeder cells (1:10 ratio) and 100 U/ml IL-2. Prior to re-stimulation CMV-pp65-specific T cells were rested in 2 U/ml IL-2 for 2 days. To exclude a possibility of residual feeders in the culture, non-T-cells were removed by magnetic depletion and subsequent enrichment using CD14 and CD4 MicroBeads (Miltenyi), respectively.

To assess the antigen-presentation capacity of untreated ILCs, autologous PBMCs were magnetically CD3 depleted (Miltenyi) and cryo-preserved. For re-stimulation CD3− PBMCs were thawed and a fraction was kept for the positive and negative stimulation controls. Rest of the cells were enriched for the CD127-expressing cells using human CD127 MicroBead Kit and loaded with CMV-pp65 protein (both Miltenyi) in the presence of 2 U/ml IL-2 overnight. Thereafter cells were harvested, surface stained and FACS sorted for separate ILC populations as described above. CMV-pp65 protein-loaded and unloaded CD3− PBMCs were sorted as control.

To analyze the antigen-presentation capacity of cytokine-treated ILCs, sorted ILC2 and ILC3-like cell populations were expanded with IL-2 ± IL-1β + IL-18 for 5–21 days (median expansion time 10 days) depending on the initial cell number and expansion rate, and loaded with CMV-pp65 protein (Miltenyi) overnight prior to stimulation. Prior to co-cultures loaded cells were purified from residual protein using CD45 MACS enrichment (Miltenyi). To ensure that detected T-cell activation is not a consequence of residual CMV-pp65 protein and self-antigen-presentation, an empty well was protein loaded and treated in the same manner as loaded cells.

ILC/CD3− PBMC-T-cell co-cultures were performed in a 1:2 ratio for 16 h. To block cytokine secretion GolgiPlug™ and GolgiStop™ (BD Biosciences) were added 4 h after the beginning of co-cultures. Subsequently the cells were harvested, stained for surface markers and intracellular cytokines (Supplementary Table 2) and analyzed as described above.

**Phospho-flow.** Phospho-flow analysis was performed according to a protocol described before[66] with minor modifications. Sorted PB ILC2 or ILC3-like cells were rested in IMDM Yssel's plus 1% NHS overnight at 37 °C. Stimulus was added and cells were stimulated for 10, 30, or 60 min at 37 °C. At the end of stimulation cells were fixed with an equal volume of 4% formaldehyde for 10 min at 37 °C. After fixation, cells were washed with FACS buffer (0.01 M PBS, 2% FCS, 2 mM EDTA) and permeabilized with 0.05% TX-100 in FACS buffer for 5 min on ice. Cells were subsequently barcoded using Pacific Blue™ Succinimidyl Ester (final concentration—1 μg/ml in DMSO plus 3x 1:3 dilution steps and one unlabeled well) and Alexa Fluor™ 488 5-SDP Ester (final concentration—0.6 μg/ml in DMSO plus 1x 1:3 dilution step and one unlabeled well) (both Thermo Fisher Scientific). Cells were incubated for 15 min at room temperature in the dark, spun down, washed, and pooled together. Pooled cells were treated with 90% methanol at −20 °C for 60 min. Afterward, cells were washed twice and stained with specific antibodies (Supplementary Table 2) for 60 min at room temperature in the dark.

**Statistics.** Statistical analysis was performed using Prism software v. 6 and v. 8 (GraphPad). Statistical difference between data sets was assessed using Mann–Whitney $U$ test (in case of unpaired measurements) and Wilcoxon matched-pairs signed rank test or Friedman test together with Dunn's multiple comparison test (for paired measurements). Correlation analysis was performed

using Pearson's correlation coefficient. Two-tailed P-values <0.05 were considered significant.

**Reporting summary**. Further information on research design is available in the Nature Research Reporting Summary linked to this article.

## Data availability

The source data underlying Figs. 1b, 1e, 2b, 2d, 3c, 4a, 4c, 4e, 5c, 6c-f and 7b and Supplementary Figs. 1c-g, 2b, 8, 10b, 14b, 15b, 16, 17a and 18a, b are provided as a Source Data file. The data that support the findings of this study are available from the corresponding authors upon reasonable request. Affymetrix microarray data generated during the current study are available on the Gene Expression Omnibus (GEO) database under the accession number GSE145626.

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

# ARTICLE

52. Mager, L. F., Wasmer, M. H., Rau, T. T. & Krebs, P. Cytokine-induced modulation of colorectal cancer. *Front Oncol.* **6**, 96 (2016).
53. Xu, L. et al. Transcriptome analysis of human colorectal cancer biopsies reveals extensive expression correlations among genes related to cell proliferation, lipid metabolism, immune response and collagen catabolism. *Oncotarget* **8**, 74703–74719 (2017).
54. Slattery, M. L. et al. The NF-kappaB signalling pathway in colorectal cancer: associations between dysregulated gene and miRNA expression. *J. Cancer Res. Clin. Oncol.* **144**, 269–283 (2018).
55. Tsushima, H. et al. High levels of transforming growth factor beta 1 in patients with colorectal cancer: association with disease progression. *Gastroenterology* **110**, 375–382 (1996).
56. Cui, G., Yuan, A., Goll, R. & Florholmen, J. IL-17A in the tumor microenvironment of the human colorectal adenoma-carcinoma sequence. *Scand. J. Gastroenterol.* **47**, 1304–1312 (2012).
57. Bernink, J. H. et al. Interleukin-12 and -23 control plasticity of CD127(+) group 1 and group 3 innate lymphoid cells in the intestinal lamina propria. *Immunity* **43**, 146–160 (2015).
58. Latz, E., Xiao, T. S. & Stutz, A. Activation and regulation of the inflammasomes. *Nat. Rev. Immunol.* **13**, 397–411 (2013).
59. Ngiow, S. F., Teng, M. W. & Smyth, M. J. A balance of interleukin-12 and -23 in cancer. *Trends Immunol.* **34**, 548–555 (2013).
60. Fenton, T. M. et al. Inflammatory cues enhance TGFbeta activation by distinct subsets of human intestinal dendritic cells via integrin alphavbeta8. *Mucosal Immunol.* **10**, 624–634 (2017).
61. Ihara, S., Hirata, Y. & Koike, K. TGF-beta in inflammatory bowel disease: a key regulator of immune cells, epithelium, and the intestinal microbiota. *J. Gastroenterol.* **52**, 777–787 (2017).
62. Carvalho, B. S. & Irizarry, R. A. A framework for oligonucleotide microarray preprocessing. *Bioinformatics* **26**, 2363–2367 (2010).
63. MacDonald, J. W. Affycoretools: Functions useful for those doing repetitive analyses with Affymetrix GeneChips. *R package version 1.0* (2008).
64. MacDonald, J. W. clariomdhumantranscriptcluster.db: Affymetrix clariomdhuman annotation data (chip clariomdhumantranscriptcluster). *R package version 8.7.0* (2017).
65. Irizarry, R. A. et al. Summaries of Affymetrix GeneChip probe level data. *Nucleic Acids Res.* **31**, e15 (2003).
66. Krutzik, P. O. & Nolan, G. P. Fluorescent cell barcoding in flow cytometry allows high-throughput drug screening and signaling profiling. *Nat. Methods* **3**, 361–368 (2006).

## Acknowledgements

We would like to thank (1) the Flow Cytometry Core Facility, Department of Medicine Huddinge, Karolinska Institutet; (2) Microscopy for this study was performed at the Live Cell Imaging facility, Karolinska Institutet, Sweden, supported by grants from the Knut and Alice Wallenberg Foundation, the Swedish Research Council, the Centre for Innovative Medicine and the Jonasson center at the Royal Institute of Technology, Sweden. (3) We also would like to thank the core facility at Novum, BEA, Bioinformatics and Expression Analysis, which is supported by the board of research at the Karolinska Institute and the research committee at the Karolinska hospital. (4) Prof. Karl-Johan Malmberg's research group (Center for Infectious Medicine, Karolinska Institutet) for kindly providing us with CMV$^+$ buffy coats; (5) Associate Prof. Jakob Michaëlsson (Center for Infectious Medicine, Karolinska Institutet) for scientific discussions and providing reagents; (6) Dr. Martin Rao for providing scientific input. The German Research Foundation (Deutsche Forschungsgemeinschaft) postdoctoral fellowship (RA 2986/1-1) to A.R. The Swedish Cancer Foundation (130396, 160664, and 170082), The Swedish Research Council (521-2013-2791), The Swedish Society for Medical Research (4-140/2014), The Swedish Foundation for Strategic Research (FFL15-0120) and the Knut and Alice Wallenberg Foundation (4-1198/2016) to J.M. EMBO long-term fellowship (ALTF 786-2013) to M.B. ERC-2013-ADG (341038) to H.S. Open access funding provided by Karolinska Institutet.

## Author contributions

A.R. formulated research questions, designed and executed the experiments, analyzed and interpreted the data and wrote the manuscript. O.S. designed and performed microscopy experiments and co-wrote and the manuscript. E.K. designed, performed and analyzed experiments. M.B. contributed to formulation of research questions, experimental design, data interpretation and critical review of the manuscript. K.P.T. performed gene expression data analysis. H.Sch. designed and optimized the phospho-flow method used in the study. A.C.G. contributed to phenotypic characterization of CRC tissue samples. L.M. provided technical assistance and scientific input. V.K. contributed to phenotypic characterization of CRC tissue samples. E.J.V. contributed to the gene expression study. N.K.B. participated in the interpretation of the microscopy data and critically revised the manuscript. U.L. formulated research questions, provided clinical samples and relevant clinical information. H.S. contributed to formulation of research questions, data interpretation and critical review of the manuscript. J.M. formulated research questions, contributed to experimental design, interpreted the data and co-wrote the manuscript.

## Competing interests

The authors declare no competing interests.
