## [Peer Review File · Nature Communications]

Reviewers' comments:

Reviewer #1 (Remarks to the Author):

The manuscript by Rao et al is a continuation of previous work of the research group demonstrating the presence of HLA-DR+ CD127+ ILC3s in human tonsils (Björklund et al. Nat. Immunol)

This interesting study starts with the observation that CD127+HLA-DR+ ILCs accumulate in colorectal cancer tissues and co-localize with T cells in healthy and tumor colon tissues. The authors further explored the capacity of ILCs to take up OVA protein and to present CMV-pp65 to Ag-specific CD4+ memory T cells. After stimulation with IL-2 and IL-1 β /IL-18 protein-loaded ILC2s and ILC3s from peripheral blood express HLA-DR and co-stimulatory molecules in vitro and induce recall responses in CMV-specific memory CD4+ T cells. The authors suggest that the upregulation of HLA-DR and co-stimulatory molecules depends on NF- κ B signalling.

In mice, IL-1 β has been shown to induce MHCII and co-stimulatory molecules in splenic but not in colonic or mLN ILC3s. The study by Rao et al. clearly adds relevant information to these studies in mouse models by demonstrating that human ILCs act as APCs for CD4+ T cells. In contrast to a previous study by Hepworth et al. in mice, the authors observed induction of HLA-DR and co-stimulatory molecule expression on intestinal (healthy or tumor) ILCs following IL-2 plus IL-1 β and IL-18 stimulation.

Major points:

The title implies that ILCs in colorectal cancer tumors are antigen presenting cells. However, data in figure 1 only support the conclusion that MHCII-expression is higher on tumor-associated ILCs as compared to healthy tissue ILCs. Unless the authors can show that ILCs isolated from tumors present Ag to CD4+ T cells and induce T cell responses, the title of the manuscript needs to be changed. Their data confirm studies by others about MHCII expression of breast and gastrointestinal tumors infiltrating ILCs (Salimi et al. BMC Cancer, 2018, 18:341). Is the enrichment of HLA-DR+ ILCs in colorectal cancer tissues a result of increased IL-1 β or IL-18 expression? The authors should measure IL-1 β , IL-18 and TGF β transcripts in tumor border and tumor tissues as well as healthy regions of their samples. In addition, data on the frequency of total ILCs in tumor tissue compared to healthy tissue are missing. Are there differences in the frequency and HLA-DR expression of total ILCs in the peripheral blood of patients vs. healthy donors? Finally, several studies have confirmed tissue-specific properties of ILC subsets. Are ILCs in tumors and control colon tissues the same or do tumor ILCs belong to a distinct subset?

Multicolor immunofluorescence stainings of colon tissues are very descriptive and need more detailed quantification and statistics in order to estimate the significance of localization of HLA-DR expressing ILCs in proximity to T cells. Although the interpretation that this may have an impact on intratumoral CD4+ T cell responses is reasonable, it is not clear whether this has an effect on tumor growth. Did the authors examine variations amongst patients, which correlate with the disease progression?

In Supplem. Fig 10 the authors show that ILCs stimulated with IL-2/IL-1 β /IL-18 significantly lose CD127. Can they exclude that the frequency of "activated" HLA-DR+ ILCs in the immunofluorescence stainings is therefore underestimated and that HLA-DR+CD127- cells are also ILCs? It would be helpful to include in situ ROR γ stainings, since ROR γ stainings of ILCs of tumor border and non-affected tissues appears to work well (Supplem. Fig 1E). In addition, ROR γ stainings should be included in the experiment of Supplem. Fig 10 to confirm the identity and purity of ILC3s. The authors should comment on their finding that after 5d culture cytokine-stimulated ILC3s express NKp44 (Supplem. Fig. 10). Are tumor-infiltrating ILCs also NKp44 positive cells? To which subset belong ROR γ -CD117-ILCs? Did they further characterize these cells?

There is a significant inhibitory effect of IL-23 on HLA-DR and co-stimulatory molecule expression of

cytokine-activated ILC3s (Fig. 4A). However, no p-Values support the conclusion that “similar to ILC3-like cells”, the combination of IL-1 β and IL-23 led to a reduced expression of these molecules on ILC2s. Statistical analysis and P-values should be added to Fig. 4C, or the text (page 10) needs to be changed.

The authors state that the PB and intestinal ILCs are comparable in terms of HLA-DR and co-stimulatory molecule expression. On the other hand they show that 2 cytokines mainly found in the intestine (IL-23 and TGF β) have a suppressive effect on the expression of these molecules. Given the fact that the cytokine microenvironment regulates HLA-DR and co-stimulatory molecule expression it is not clear why PB ILCs and intestinal ILCs express comparable levels of these molecules. Did the authors investigate other co-stimulatory/inhibitory receptors, activation marker, integrins and chemokine receptors? Is the transcriptional profile similar to tonsil ILCs?

Ex vivo PB ILCs were shown to take up and process DQ-OVA, albeit less efficient than classical APCs (Fig. 2). Is the efficiency of DQ-OVA uptake and processing increased when ILCs are stimulated with IL-2/IL-1 β /IL-18 before Ag-loading?

In their APC assay (Fig. 3 and 6) only cytokine responses are measured as readout for specific CD4+ T cell responses. Considering the finding that cognate recognition of Ag-presenting mouse intestinal ILCs was shown to inhibit CD4+ T cell proliferation it would be important to know whether human CD4+ T cells can proliferate in response to Ag-presenting ILCs. Did the authors test whether the effect of cytokine treatment on HLA-DR and co-stimulatory molecule expression was sufficient to induce specific responses of naïve CD4+ T cells? In the cell isolation and sorting protocol it is not explained how the authors excluded DCs. This is important in order to avoid DC contamination in the APC assays.

Why does IL-23 only have a suppressive effect on HLA-DR and co-stimulatory molecule expression at day 5? Did the authors test whether IL-23 has an influence on cell viability? The effect of TGF β and IL-23 +/- IL-1 β /IL18 should also be tested in the APC assay with Ag loaded ILCs and T cells.

IKK/NF κ B inhibitor BAY 11-7082 has multiple other targets (PMIDs 23441730, 23578302, 22745523). Therefore, the effect of the inhibitor could have alternative reasons. Additional verification of the role of NF κ B is required.

Figure 6 is confusing. Do CD3-, ILC2s and ILC3s have the same efficiency as APCs despite a significant difference in HLA DR expression of ILC3s and ILC2s (Fig. 4A, C), in co-stimulatory molecule expression of ILC3s and ILC2s (Fig. 4B,D), and in processing capacity of DCs and ILCs (Fig. 2)? Why is the % T cell response similar when CMV-pp65 protein-loaded CD3- cells or cytokine-stimulated ILC2s or ILC3s are used (Fig. 6C)? How does this fit to the statement on page 14 and data in Fig. 6E that the magnitude of the observed T-cell response correlated with the HLA-DR expression levels on the antigen-presenting ILC populations in co-culture?

Fig. 6C: The authors should compare cells stimulated with IL-2 alone or with IL-2/IL-1 β or with IL-2/IL-1 β /IL-18 and provided p-values. This is more relevant in terms of estimating the effect of inflammatory cytokines on T-cell-stimulatory properties of ILCs.

Fig. 6D: What is the meaning of the colored slices?

Minor points:

Fig 1B: Frequency of HLA-DR+ ILCs is shown. How was the gating done?

The authors state on page 9 that there is a weak upregulation of HLA-DR after 24h cytokine treatment. This is not evident in the Supplem. Fig. 6. The text should be corrected accordingly. Why is there a decrease of CD70 after cytokine treatment?

Supplem. Fig. 8: Amongst others, the expression of inhibitory molecules such as PD-L1 and PD-L2 on sort-purified PB ILC3-like cells and ILC2s after IL-2 plus IL-1 β stimulation is shown. Does IL-23 and TGF β exposure have an effect on the expression of inhibitory molecules?

In the legend of Fig. 6, slice instead of slide should be written.

Reviewer #2 (Remarks to the Author):

In this manuscript, Rao et al. examine the antigen processing and presenting capabilities of human ILC2 and ILC3-like cells, mainly from peripheral circulation but also from colon tissue. The authors observe that peripheral ILCs do not act as MHC class II presenting APCs directly ex vivo but that they can be induced to present a CMV-pp65 peptide from recombinant protein to memory CD4 T cells following prolonged exposure to cytokines (IL-2 +/- IL-1 β +/- IL-18). This treatment upregulates class II, and in some cases CD70, CD80 and CD86. Use of chemical inhibitors suggest that upregulation is NF κ B- but not STAT1-dependent. Finally, the authors show that intestinal ILCs upregulate class II, CD70, CD80 and CD86 in response to prolonged exposure to the same three cytokines.

The central question asked by the authors is an important one but enthusiasm is reduced by several considerations:

- There are several apparent contradictions in the manuscript that create confusion about key points:
 - o The authors attach much significance to the expression of co-stimulatory molecules in conferring APC capabilities but show that IL-2 treatment alone induces memory CD4+ T cell stimulating capability without inducing expressing of co-stimulatory molecules.
 - o A core manipulation is prolonged exposure of ILC populations to cytokines ex vivo, but it is not clear that these treatments reflect in vivo conditions and the authors even concede this in the Discussion (Lines 367-369).
 - o The authors show no presentation by ILCs directly ex vivo without prolonged cytokine treatment but state in the discussion (Lines 358-360), that it could be different in vivo due to key differences in T cell composition, microenvironment, etc...
 - o Related to this, the title includes the phrase "cytokine microenvironment" but this microenvironments are not directly examined.
- Additional concerns:
 - o Other than cytokine responsiveness and lack of co-stimulatory molecules, the peripheral and intestinal ILCs are not functionally connected. Are there data showing that they have similar transcriptional programs, for instance?
 - o Are there conditions in vivo where ILCs are shown to express co-stimulatory molecules? I looked for this information in the manuscript and did not find it. If I did not miss it, this is important to mention or show. Otherwise, the observations risk being only in vitro phenomena.
 - o On this note, for colonic ILCs the authors show upregulation of class II and co-stimulatory molecules but actual antigen presentation (even with peptide) is not shown. In general, the authors lead with surface markers and then follow, almost as an afterthought, with actual antigen presentation, which does not make sense to me in light of the previous points.
 - o The work is largely descriptive. An exception is the investigation of signaling requirements mentioned above. There are two problems with these studies (Figure 5). First, the authors do not address the possibility that reduced expression of the various proteins caused by BAY11-7082 is due to toxicity, even though toxicity to ILC2 cells was noted (lines 276-277). Second, the authors do not show that effective doses of Flurabine phosphate were utilized. (Is there a STAT1-dependent process in ILCs that could have served as a positive control?)
 - o For the imaging experiments in Figure 1, co-localization of class II-positive cells and CD4 cells does not mean activation. An analysis that shows actual activation in situ would substantially strengthen the work. Related to this, the authors observe that ILCs, HLA-DRhiCD45+ and T cells frequently co-

localize, suggesting "mutual regulatory mechanisms". (Lines 129-131). Are the authors suggesting a processing and presentation function for ILCs in this context?

o Inhibitor experiments were carried out following cytokine exposure for 1 hour at most when other experiments were carried out after several days of cytokine exposure, shorter incubations not leading to functional differences. Is the several-days exposure to the three cytokines realistic? Could this happen in vivo?

o Figure 7: The data for no treatment ("Ex vivo") are not provided for the tumor border and tumor.

o The p values in Figure 1B are really 0.003, 0.005 and 0.002?

Reviewer #3 (Remarks to the Author):

In this manuscript, Rao et al. investigate the antigen presenting capacities of human ILCs and how cytokines regulate this function in vitro. They show that ILCs present in colorectal tumors (or their vicinity) express elevated levels of HLA-DR although lacking co-stimulatory molecules on their surface. Blood ILCs were able to uptake and process ovalbumin and when stimulated by IL-1b demonstrate features of antigen presenting cells. These cells were subsequently able to induce recall responses in memory T cells. Overall, the authors' main conclusions are supported by their presented data which appear to be mostly novel. Experiments are well-conducted. This manuscript confirms and complements the findings of Ohne et al (NatImmunol 2016), describing a role for IL-1b in ILC2 maturation and function. However, there are a few concerns with the manuscript in its current form as follows :

1. Authors should cite the paper of Ohne et al (NatImmunol 2016), since it is complementary to their study, especially because they are showing cytokines production and transcriptome changes in ILC2 stimulated with IL-1b.

2. Throughout the manuscript, the authors state that both IL-1b and IL-18 induce the antigen-presenting phenotype observed in ILCs. The data don't support this conclusion as only IL-1b is inducing upregulation of MHCII and costimulatory molecules. IL-18 alone doesn't seem to have any role and doesn't show any additive or synergistic effect in combination with IL-1b except slightly on ILC2s (Figures 4, 5, 6). It is therefore misleading to state that IL-18 has a role in this ILC function since it doesn't seem to have a role on ILC3 and a weak role on ILC2. The authors should correct this in the manuscript and be less affirmative on the role of IL-18 (lines 240, 284 for example).

3. In the abstract, line 28, « intestinal and peripheral blood ILCs... » is also misleading, since only blood ILCs were actually tested ex vivo for APC characteristics in several cytokine conditions. Only Figure 1 shows intestinal ILCs.

4. In Figure 1A, the CD117+ and CD117- subsets are difficult to distinguish. The authors should modify the y axis so the two populations become distinct. This should be feasible with analyzing softwares such as diva or flowjo or by adding a cross on the dot plot.

5. Concerning figure 1C, D, the authors observe that ILC co-localize with T cells, with APCs or both (« frequently observed » line 129). It would be interesting and would add to the manuscript if the authors could quantify the different interactions in the several anatomical regions. If ILCs are more often observed in contact with a T cell or with Tcell+APC than alone, that would strengthen the proposed hypothesis that ILC can present antigens and reactivate memory T cells.

6. In figure 2C, D, the authors do not show data on ILC2, even though Fig 2A shows ILC2. This information on whether ILC2 are able to process antigen (DQ-OVA here) is crucially missing, especially regarding the conclusions of the manuscript emphasizing on the potential of ILC2 to act as antigen-presenting cells. Also, as it is now, the manuscript states that « degradation is equally efficient in all ILC subsets » (line 158) which is untrue since data on ILC2 is missing.

7. In figure 4, the authors should show upregulation of HLA and costimulatory molecules on professional APCs such as Lin+HLA-DRhi cells (as in Figure 2) in parallel to the data on ILC3 and ILC2. This would greatly help to evaluate the extend of the upregulation of these molecules and the potential « APCness » of ILCs. Figure 2 clearly demonstrates that even though ILCs seem to be able to act as APCs, they are way less efficient at processing DQ-OVA than professional APCs. The parallel

should be made for levels of HLA and costim molecules as well.

8. Figure 4 shows that ILC2 respond to IL-23. Are there other studies that describe ILC2 responding to IL-23 ? if so, authors should provide references. Isn't it possible that these cells are ILC2 that converted to an ILC3 phenotype ? The authors mention that these stimulated ILC2 and ILC3like cells do not express NKG2A or CD94, ruling out that these cells acquire an ILC1/NK-like phenotype.

However, supp Figure 10 shows a strong upregulation of NKp44 on ILC3 cells after cytokine stimulation, that could suggest that these cells are converting to an NCR+ ILC3 or ILC1-like phenotype. Ohne et al. (NatImmunol 2016) show that IL-1b-stimulated ILC2 upregulate Tbet, Eomes and several cytokine receptors and differentiate into ILC1-like cells. How do the authors interpret their results in view of the Ohne study ? Staining for Tbet, Eomes, RORgt for exemple would add greatly to the manuscript and to understand the complex role of cytokines in ILC function and plasticity.

9. The conclusion stated on line 284 that IL-1b and IL-18 stimulation leads to NFkb signaling needed for optimal HLA and costimulatory molecule expression is not supported by the data in the figure 5 which doesn't show HLA and costimulatory molecule expression on cells stimulated by IL-18 (Figure 5C only shows data for IL-1b). The authors should correct this.

Reviewers' comments:

Reviewer #1 (Remarks to the Author):

The manuscript by Rao et al is a continuation of previous work of the research group demonstrating the presence of HLA-DR+ CD127+ ILC3s in human tonsils (Björklund et al. Nat. Immunol)

This interesting study starts with the observation that CD127+HLA-DR+ ILCs accumulate in colorectal cancer tissues and co-localize with T cells in healthy and tumor colon tissues. The authors further explored the capacity of ILCs to take up OVA protein and to present CMV-pp65 to Ag-specific CD4+ memory T cells. After stimulation with IL-2 and IL-1 β /IL-18 protein-loaded ILC2s and ILC3s from peripheral blood express HLA-DR and co-stimulatory molecules in vitro and induce recall responses in CMV-specific memory CD4+ T cells. The authors suggest that the upregulation of HLA-DR and co-stimulatory molecules depends on NF- κ B signalling.

In mice, IL-1 β has been shown to induce MHCII and co-stimulatory molecules in splenic but not in colonic or mLN ILC3s. The study by Rao et al. clearly adds relevant information to these studies in mouse models by demonstrating that human ILCs act as APCs for CD4+ T cells. In contrast to a previous study by Hepworth et al. in mice, the authors observed induction of HLA-DR and co-stimulatory molecule expression on intestinal (healthy or tumor) ILCs following IL-2 plus IL-1 β and IL-18 stimulation.

Major points:

1. The title implies that ILCs in colorectal cancer tumors are antigen presenting cells. However, data in figure 1 only support the conclusion that MHCII-expression is higher on tumor-associated ILCs as compared to healthy tissue ILCs. Unless the authors can show that ILCs isolated from tumors present Ag to CD4+ T cells and induce T cell responses, the title of the manuscript needs to be changed.

Response: We agree with the reviewer that the title needs to be changed. With the new insight generated by the revision experiments, we have now changed the title to: The cytokine microenvironment regulates the antigen-presenting characteristics of circulating and tissue-resident intestinal ILCs in humans. We believe that this title better reflects the data presented.

2. Their data confirm studies by others about MHCII expression of breast and gastrointestinal tumors infiltrating ILCs (Salimi et al. BMC Cancer, 2018, 18:341). Is the enrichment of HLA-DR+ ILCs in colorectal cancer tissues a result of increased IL-

1 β or IL-18 expression? The authors should measure IL-1 β , IL-18 and TGF β transcripts in tumor border and tumor tissues as well as healthy regions of their samples.

Response: We agree that this question is very interesting but due to the translational nature of the data, causality is difficult to address. Nevertheless, we performed affymetrix microarray analysis of whole-tissue fragments of non-affected, tumor-border and tumor tissue of three CRC patients. As presented in supplementary figure 8, we can indeed detect mRNA of all the cytokines investigated in the study (*IL1B*, *IL18*, *IL23A*, *IFNG*, *TGFB1* and *TGFB3*) in both tumor and non-tumorous areas, making the mechanisms that we describe *in vitro* plausible also *in vivo*. Although the aforementioned cytokines are all implicated in CRC (West *et al.*, Nature Reviews Immunology, 2015), our microarray data indicate that the enrichment of HLA-DR⁺ ILCs in CRC tumors is not only explained by increased transcription of these cytokines. Other larger studies have however reported increased expression of transcripts of *IL1B*, *IL23A* and *TGFB1*. This is discussed on page 18-19.

3. In addition, data on the frequency of total ILCs in tumor tissue compared to healthy tissue are missing.

Response: We have added statistics on the frequency of total ILCs among CD45⁺ lymphocytes and CD45⁺CD3⁻ lymphocytes to supplementary figure 1E. The data shows that there is no difference in the total frequency of ILCs in tumor tissue compared to healthy tissue. This is discussed on page 5.

4. Are there differences in the frequency and HLA-DR expression of total ILCs in the peripheral blood of patients vs. healthy donors?

Response: In the edited supplementary figure 1F, we compared the frequency of peripheral blood HLA-DR⁺ ILCs (% of total ILCs) between the CRC patients and healthy donors and found no statistical differences between the two groups. This is discussed on page 5.

5. Finally, several studies have confirmed tissue-specific properties of ILC subsets. Are ILCs in tumors and control colon tissues the same or do tumor ILCs belong to a distinct subset?

Response: We agree with the reviewer that this is an interesting question to address. To understand more about the phenotype and function of tumor vs. non-affected colon tissue we analyzed markers associated with tissue-residency and maturation of ILCs, including CD69, NKp44 and ROR γ t (supplementary figure 1G and 2A-B). The results show that while circulating blood ILCs are CD69⁻, non-affected colon tissue

ILCs are predominantly tissue-resident mature $ROR\gamma t^+NKp44^+CD69^+$ ILC3. Interestingly, tumor-associated ILCs express significantly less CD69, NKp44 and $ROR\gamma t$, suggesting that they are less tissue resident and display a less pronounced ILC3-phenotype than non-affected colon tissue. This might indicate that circulating ILCs are infiltrating the highly vascularized tumor tissue. Of note, HLA-DR expression on tumor ILCs is seen on both $CD69^+$ and $CD69^-$ cells (Supplementary Fig. 1H), as well as on $NKp44^+$ and $NKp44^-$ ILCs (data not shown). This is discussed on pages 6 and 19.

6. Multicolor immunofluorescence stainings of colon tissues are very descriptive and need more detailed quantification and statistics in order to estimate the significance of localization of HLA-DR expressing ILCs in proximity to T cells.

Response: We have now performed a quantitative assessment of the immunofluorescence data that we have added to Figure 1E. We observe that 52% and 38% of all ILCs are located in immediate proximity to T cells in non-affected and tumor-border tissue, respectively. These data emphasize the fact that ILC-T cell interactions are likely to occur. This is discussed on page 7.

7. Although the interpretation that this may have an impact on intratumoral $CD4^+$ T cell responses is reasonable, it is not clear whether this has an effect on tumor growth. Did the authors examine variations amongst patients, which correlate with the disease progression?

Response: Although this is a very interesting question, since this is a prospective study involving analysis of fresh tumor tissues, not enough time has passed to address disease progression. We did however analyze if there were differences in the frequencies of intratumoral HLA-DR⁺ ILCs depending on the CRC TNM score i.e. the size of the tumor, lymph node involvement and metastasis. We found no significant correlations but still decided to color-code the data points in figure 1B on the basis of the cancer stage score. This is discussed on page 5.

8. In Supplem. Fig 10 the authors show that ILCs stimulated with IL-2/IL-1 β /IL-18 significantly lose CD127. Can they exclude that the frequency of "activated" HLA-DR⁺ ILCs in the immunofluorescence stainings is therefore underestimated and that HLA-DR⁺CD127⁻ cells are also ILCs? It would be helpful to include in situ $ROR\gamma$ stainings, since $ROR\gamma$ stainings of ILCs of tumor border and non-affected tissues appears to work well (Supplem. Fig 1E).

Response: We agree with the reviewer that activated ILCs might down-regulate CD127 and therefore we might underestimate the frequency of HLA-DR⁺ ILCs. Unfortunately we currently have no solution to this issue as we also observe that

ROR γ t is down-regulated in the tumor tissue (supplementary figure 2A, B). Hence, given the known difficulties in staining for ROR γ t in humans, it is unlikely to be possible to identify tumor ILCs based on this transcription factor.

9. In addition, ROR γ stainings should be included in the experiment of Supplem. Fig 10 to confirm the identity and purity of ILC3s. The authors should comment on their finding that after 5d culture cytokine-stimulated ILC3s express NKp44 (Supplem. Fig. 10). Are tumor-infiltrating ILCs also NKp44 positive cells?

To which subset belong ROR γ -CD117-ILCs? Did they further characterize these cells?

Response: We have now added transcription factor stainings of *in vitro* cultured ILC3-like cells and ILC2 (Supplementary figure 13D). IL-1 β is known to induce plasticity of human ILCs (Bal *et al.*, Nat Immunol, 2016; Ohne *et al.*, Nat Immunol, 2016; Golebski *et al.*, Nat Comm, 2019; Mazzurana *et al.*, EJI, 2019, Bernink, Nature Immunol, 2019; Hochdörfer, EJI, 2019) and in line with this, we observe that IL-1 β -cultured ILC3-like cells maintain ROR γ t and intermediate T-bet expression while upregulating GATA3. IL-1 β -cultured ILC2 upregulate ROR γ t and T-bet, while exhibiting high GATA3 expression. Eomes was not detected in any condition. However, although IL-1 β alters the transcription factor expression, it is not enough for full ILC3-to-ILC1 (Mazzurana *et al.*, EJI, 2019) or ILC2-to-ILC1/3 (Bal *et al.*, Nat Immunol, 2016; Ohne *et al.*, Nat Immunol, 2016; Golebski *et al.*, Nat Comm, 2019; Bernink, Nature Immunol, 2019; Hochdorfer, EJI, 2019) transdifferentiation. For ILC3/ILC2-to-ILC1 differentiation, IL-12 is needed, and for ILC2-to-ILC3 differentiation IL-23+TGF- β is required. Hence, although IL-1 β causes ILCs to be more responsive to lineage-polarizing cytokines, it is not enough for full transdifferentiation to other ILC lineages. This fits with the conclusion by Ohne *et al.*, Nat Immunol, 2016 that IL-1 β potentiates activation and plasticity and IL-12 acts as a switch in ILC2-to-ILC1 transdifferentiation. This is discussed on pages 12 and 19-20.

Regarding NKp44, we refer to our answer to point 5 and Supplementary Fig. 1G. Importantly, HLA-DR expression is seen on both NKp44⁺ and NKp44⁻ ILCs in the tumor (data not shown).

Regarding CD117⁻ ILCs, in the previous supplementary fig 1E, these cells were used as a negative control for ROR γ t as CD117⁻ ILCs in tissues have been shown to harbor ILC1 (Bernink *et al.*, Nat Immunol, 2013). However, PB CD117⁻ ILCs remain largely uncharacterized and likely constitute a highly heterogeneous population (Roan *et al.*, J Immunol, 2016). We initially addressed the APC-function of these cells but they proved difficult to maintain in culture. Hence, from Fig. 4 and onwards, CD117⁻ ILCs were left out of the manuscript. Performing a full characterization of these cells is beyond the scope of the current study.

10. There is a significant inhibitory effect of IL-23 on HLA-DR and co-stimulatory

molecule expression of cytokine-activated ILC3s (Fig. 4A). However, no p-Values support the conclusion that “similar to ILC3-like cells”, the combination of IL-1 β and IL-23 led to a reduced expression of these molecules on ILC2s. Statistical analysis and P-values should be added to Fig. 4C, or the text (page 10) needs to be changed.

Response: We thank the reviewer for noticing this mistake. There is no significant effect of IL-23 on IL-2+IL-1 β -induced HLA-DR- or co-stimulatory-molecule expression. We have now removed this statement in the text.

11. The authors state that the PB and intestinal ILCs are comparable in terms of HLA-DR and co-stimulatory molecule expression. On the other hand they show that 2 cytokines mainly found in the intestine (IL-23 and TGF β) have a suppressive effect on the expression of these molecules. Given the fact that the cytokine microenvironment regulates HLA-DR and co-stimulatory molecule expression it is not clear why PB ILCs and intestinal ILCs express comparable levels of these molecules. Did the authors investigate other co-stimulatory/inhibitory receptors, activation marker, integrins and chemokine receptors? Is the transcriptional profile similar to tonsil ILCs?

Response: We are thankful to the reviewer for noticing that we did not formulate ourselves clear enough. We have now changed the text on page 8 related to blood vs. gut HLA-DR expression. Although ILCs in both tissues can express HLA-DR, intestinal ILC more frequently express HLA-DR (43-62%; Figure 1B) than blood ILCs (17%; Supplemental figure 1F). This could indeed be related to the tissue-microenvironment. We agree that IL-23 and TGF β protein levels are likely to be higher in the intestine than in the blood. However, given that ILCs are often located close to HLA-DR^{high} DC/macrophage-like cells in the intestine (Figure 1C-E), intestinal ILCs are also likely exposed to higher levels of IL-1 β than blood ILC. Hence, the expression of HLA-DR on intestinal ILCs is the net-effect of factors that promote and inhibit the APC-features of the ILCs. However, since ILCs in neither blood nor gut (with the exception of three tumor cases displayed in supplementary figure 6) expressed any of the analyzed co-stimulatory molecules (CD80, CD86, CD70, PDL1, PDL2, 4-1BBL, OX40L, ICOSL and CD30L), we decided to use HLA-DR⁺ ILCs from blood as a model system to understand the mechanisms regulating the antigen-presenting capacity of these cells.

Although we agree with the reviewer that a transcriptional comparison of blood, gut and tonsil ILCs would be very valuable, it is beyond the scope of the current paper, which consists of 7 main and 19 supplementary figures.

12. Ex vivo PB ILCs were shown to take up and process DQ-OVA, albeit less efficient

than classical APCs (Fig. 2). Is the efficiency of DQ-OVA uptake and processing increased when ILCs are stimulated with IL-2/IL-1 β /IL-18 before Ag-loading?

Response: This is an interesting question raised by the reviewer and we addressed this in the new supplementary figure 18. Whereas IL-2+IL-1 β stimulation increases the uptake and processing of DQ-OVA in ILC2 (suppl fig 18A), it has no effect in this respect on ILC3-like cells (suppl fig 18B). DQ-OVA MFI is also not affected by the presence of TGF- β . Hence, we conclude that the effect of IL-1 β and TGF- β on the antigen-presenting capacity of ILCs is not mediated through altered capacity to take up and process proteins. This is discussed on page 17.

13. In their APC assay (Fig. 3 and 6) only cytokine responses are measured as readout for specific CD4⁺ T cell responses. Considering the finding that cognate recognition of Ag-presenting mouse intestinal ILCs was shown to inhibit CD4⁺ T cell proliferation it would be important to know whether human CD4⁺ T cells can proliferate in response to Ag-presenting ILCs.

Response: The role for ILCs in inducing proliferation of antigen-specific T cells in mice is still controversial. Whereas Hepworth et al (Nature 2013) showed reduced T cell proliferation in response to intestinal ILCs, von Burg *et al* (PNAS 2014) reported increased proliferation of naïve antigen-specific T cells by splenic ILCs, which was promoted by IL-1 β . We now extend the latter observation by developing an extensive protocol to assess the APC-function of human ILCs and their capacity to induce cytokine-production in antigen-specific memory T cells, which was not done in the paper from von Burg *et al*. Given the similarities in the effects of IL-1 β on mouse (von Burg *et al*) and human (the current paper) ILCs, we expect that also in humans, IL-1 β -activation of human ILCs would render the ILCs capable of inducing proliferation of CMV-specific memory CD4⁺ T cells, as these cells are highly proliferative in our expansion assay. Assessing proliferation would have required another set of experiments and given the extent of the revision, the time frame given, we had to prioritize the other questions of the reviewer.

14. Did the authors test whether the effect of cytokine treatment on HLA-DR and co-stimulatory molecule expression was sufficient to induce specific responses of naïve CD4⁺ T cells?

Response: This experiment is only possible in mice, which transgenically express TCRs of only one specificity e.g. OVA. In humans this is not possible since even the frequency of CMV-specific memory T cells is only maximally 1% of the total CD4⁺ T cell pool. Moreover, it has been demonstrated in mice that out of several billion cells only approximately 100 are naïve T cells that display the specificity for a single

specific peptide-MHC complex (Obar et al. Immunity, 2008; Moon et al. Immunity, 2007). Hence, the frequency of any naïve antigen-specific CD4⁺ T cell would be too low to be detected in the donor blood, preventing the possibility of addressing this question.

15. In the cell isolation and sorting protocol it is not explained how the authors excluded DCs. This is important in order to avoid DC contamination in the APC assays.

Response: We used DC- and monocyte markers in the lineage antibody cocktail to ensure that these cell types are not contaminating the sorted ILC populations. The mix includes CD14, CD1a, CD123, BDCA2 and FcER1 as seen in supplementary table II. In supplementary figure 2C we also show that CD127⁺ cells isolated from the colon do not express any of the lineage markers. We also show that expanded ILCs do not express CD40 (Supplementary figure 17B), in sharp contrast to CD3⁻ PBMC, which contains professional antigen-presenting cells such as B cells (main figure 3).

16. Why does IL-23 only have a suppressive effect on HLA-DR and co-stimulatory molecule expression at day 5? Did the authors test whether IL-23 has an influence on cell viability? The effect of TGFβ and IL-23 +/- IL-1β/IL18 should also be tested in the APC assay with Ag loaded ILCs and T cells.

Response: We analyzed viability of IL-23-stimulated ILC3-like cells after 5-day cultures and noticed no effects on viability as compared to the condition without IL-23. Please see the figure below for the reviewer's inspection only (DCM=dead cell marker):

Since the effects of IL-18 and IL-23 are much less drastic than the effects of IL-1β and TGF-β, we had to focus on these two latter cytokines given that the CMV-assay that we developed is extremely time- and resource demanding. As the reviewer

suggested, we did perform experiments to assess if TGF- β indeed suppress the antigen-presenting capacity of IL-1 β -stimulated ILCs. In line with the suppression of HLA-DR and co-stimulatory molecule expression by TGF- β , this cytokine as expected reduced the capacity of IL-1 β -stimulated ILCs to induce a response in CMV-specific memory T cells (new figure 6F).

17. IKK/NF κ B inhibitor BAY 11-7082 has multiple other targets (PMIDs 23441730, 23578302, 22745523). Therefore, the effect of the inhibitor could have alternative reasons. Additional verification of the role of NF κ B is required.

Response: We verified the data in figure 5C using another inhibitor, BMS-345541 that has been reported as highly specific in mice (PMID: 12403772) and verified in several human cancer settings (PMID: 16467110 and 22713244). These new data are added to supplementary figure 16 and discussed on page 14. The viability of the cells after BMS-treatment is given below for the reviewer's inspection only. Moreover, non-viable cells have been excluded from the final analysis (supplementary figure 16) based on CD45 and Dead Cell Marker staining. We are now emphasizing this in the method section (page 26).

18. Figure 6 is confusing. Do CD3-, ILC2s and ILC3s have the same efficiency as APCs despite a significant difference in HLA DR expression of ILC3s and ILC2s (Fig. 4A, C), in co-stimulatory molecule expression of ILC3s and ILC2s (Fig. 4B,D), and in processing capacity of DCs and ILCs (Fig. 2)?

Response: The ILCs are indeed potent antigen-presenting cells when pre-cultured with IL-1 β . The CD3 $^-$ PBMC served as a positive control in this experiment but their capacity might be reduced, as they had to be frozen and thawed in order for the protocol to work (see main figure 3 and 6). Hence, the CD4 $^+$ T cell response induced by CMV-loaded CD3 $^-$ PBMC and ILCs should not be directly compared but rather, the CD3 $^-$ PBMC serve as a positive control in this advanced experiment. We are now emphasizing this on page 16.

19. Why is the % T cell response similar when CMV-pp65 protein-loaded CD3 $^-$ cells or cytokine-stimulated ILC2s or ILC3s are used (Fig. 6C)? How does this fit to the statement on page 14 and data in Fig. 6E that the magnitude of the observed T-cell response correlated with the HLA-DR expression levels on the antigen-presenting ILC populations in co-culture?

Response: Regarding the antigen-presenting capacity of CD3 $^-$ cells, we refer to our response to question 18. Regarding the correlation between HLA-DR expression and antigen-presenting capacity of ILCs, it does suggest that higher HLA-DR expression leads to higher antigen-presenting capacity. However, it does not exclude the possibility that higher HLA-DR expression is associated with another unknown effect that enhances the APC-function of ILCs. In addition, the HLA-DR expression is very high on both IL-1 β -stimulated ILCs (MFI 10 4 , see fig 4B) and CD3 $^-$ PBMC (MFI 10 4 -10 5 , see fig 3D). Given these circumstances, we are not surprised that cytokine-stimulated ILCs are good antigen-presenting cells, to an extent that is comparable to frozen and thawed CD3 $^-$ PBMC with comparable HLA-DR expression.

20. Fig. 6C: The authors should compare cells stimulated with IL-2 alone or with IL-2/IL-1 β or with IL-2/IL-1 β /IL-18 and provided p-values. This is more relevant in terms of estimating the effect of inflammatory cytokines on T-cell-stimulatory properties of ILCs.

Response: We added two additional paired data points to figure 6C. We now show that there is a significant increase in the antigen-presenting capacity of ILC3-like cells cultured with IL-2 alone as compared to with IL-2+IL-1 β . For the other IL-2 vs. IL-2+IL-1 β and IL-2+IL-1 β +IL-18 comparisons we do not have enough paired data points to perform the statistical test.

21. Fig. 6D: What is the meaning of the colored slices?

Response: We have clarified the figure legend related to figure 6D.

Minor points:

22. Fig 1B: Frequency of HLA-DR+ ILCs is shown. How was the gating done?

Response: Please see figure 1 and supplementary figure 1.

23. The authors state on page 9 that there is a weak upregulation of HLA-DR after 24h cytokine treatment. This is not evident in the Supplem. Fig. 6. The text should be corrected accordingly. Why is there are decrease of CD70 after cytokine treatment?

Response: We agree with the reviewer and have removed the text regarding supplementary figure 9 (previous supplementary fig 6). Regarding CD70 expression on ILC2 after 24 hours, all of the conditions are negative for CD70 staining. The IL-2+IL-1 β +IL-18 condition shows a lower background for unknown reason, but it does not mean that CD70 is downregulated. None of the ILC2s stain positive for CD70 compared to e.g. the ILC3-like cells in the same figure.

24. Supplem. Fig. 8: Amongst others, the expression of inhibitory molecules such as PD-L1 and PD-L2 on sort-purified PB ILC3-like cells and ILC2s after IL-2 plus IL-1 β stimulation is shown. Does IL-23 and TGF β exposure have an effect on the expression of inhibitory molecules?

Response: We analyzed this and there is no expression of PDL1 or PDL2 on ILC3-like cells exposed to IL-23 and/or TGF β . This data is now presented in Supplementary Fig. 14

25. In the legend of Fig. 6, slice instead of slide should be written.

Response: We thank the reviewer for noticing this mistake. We now have changed this.

Reviewer #2 (Remarks to the Author):

In this manuscript, Rao et al. examine the antigen processing and presenting capabilities of human ILC2 and ILC3-like cells, mainly from peripheral circulation but also from colon tissue. The authors observe that peripheral ILCs do not act as MHC class II presenting APCs directly ex vivo but that they can be induced to present a CMV-pp65 peptide from recombinant protein to memory CD4 T cells following prolonged exposure to cytokines (IL-2 +/- IL-1 β +/- IL-18). This treatment upregulates class II, and in some cases CD70, CD80 and CD86. Use of chemical inhibitors suggest that upregulation is NF κ B- but not STAT1-dependent. Finally,

the authors show that intestinal ILCs upregulate class II, CD70, CD80 and CD86 in response to prolonged exposure to the same three cytokines.

The central question asked by the authors is an important one but enthusiasm is reduced by several considerations:

There are several apparent contradictions in the manuscript that create confusion about key points:

1. The authors attach much significance to the expression of co-stimulatory molecules in conferring APC capabilities but show that IL-2 treatment alone induces memory CD4+ T cell stimulating capability without inducing expressing of co-stimulatory molecules.

Response: We agree with the reviewer. Although IL-1 β -stimulation leads to increased antigen-presenting capacity of ILCs, it seems as co-stimulatory molecule expression is not completely necessary for re-activation of CMV-specific memory T cells, which has also been described by others (London et al, Journal of Immunology, 2000; Berard et al., Immunology, 2002). We did discuss this in the manuscript on page 21, and have now tried to be even more clear on this point on page 16 of the results section and page 22 of the discussion.

2. A core manipulation is prolonged exposure of ILC populations to cytokines *ex vivo*, but it is not clear that these treatments reflect *in vivo* conditions and the authors even concede this in the Discussion (Lines 367-369).

Response: We understand the reviewer's position but this is inherent to studies in humans where we cannot do *in vivo* experiments. However, we do show five key pieces of *in vivo/ex vivo* data that support the notion that our *in vitro* data have *in vivo* relevance.

1. We show that ILCs are often located in close proximity to T cells in non-affected and tumorous colon *in vivo* (figure 1), making ILC-T cell interactions likely to occur. Importantly, these ILC-T cell interactions seems to often happen close to a HLA-DR^{high} DC/macrophage-like cell which are possible sources of the cytokines that we examined in the study: IL-1 β , IL-18, IL-23 and TGF- β .
2. To better understand the *in vivo* situation and the intestinal microenvironment, we also performed affymetrix microarray analysis of whole-tissue fragments of non-affected, tumor-border and tumor tissue of three CRC patients. As presented in supplementary figure 8, we can indeed

detect mRNA of all the cytokines investigated in the study (*IL1B*, *IL18*, *IL23A*, *IFNG*, *TGFB1* and *TGFB3*), making the mechanisms that we describe *in vitro* plausible also *in vivo*. Although the aforementioned cytokines are all implicated in CRC (West et al., Nature Reviews Immunology, 2015), our microarray data indicate that the enrichment of HLA-DR⁺ ILCs in CRC tumors is not only explained by increased transcription of these cytokines. Other larger studies have however reported increased expression of transcripts of *IL1B*, *IL23A* and *TGFB1* (*IL1B* increased in CRC - PMID: 29188362, 29088818; *IL23A* - PMID: 29088818; *TGFB1* increased in CRC – PMID: 8566583). This is now discussed on pages 18-19.

3. We detected CD86 expression on ILCs in tumor tissues of three CRC patients (now shown in supplementary figure 5), indicating that co-stimulatory molecules can be expressed on ILCs *in vivo*.
4. Moreover, we show that *ex vivo* isolated intestinal and intratumoral ILCs respond to IL-1 β & IL-18 by upregulating HLA-DR and co-stimulatory molecules (figure 7). These data show that the antigen-presenting capacity of ILCs could be harnessed for improved CD4⁺ T cell immunity, which we also highlight in the discussion and abstract.
5. We show that there is a correlation between the level of HLA-DR expression on ILCs, and their capacity to induce cytokine expression in CMV-specific CD4⁺ T cells *in vitro*. We also show that this is not necessarily due to co-stimulatory molecule expression, as ILCs stimulated with IL-2 alone (which do not express co-stimulatory molecules) can cause re-call responses in CD4⁺ memory T cells. These data, together with the fact that intestinal ILCs express significantly more HLA-DR than blood ILCs *ex vivo*, raises the possibility that intestinal ILCs, in contrast to *ex vivo* blood ILCs, might in fact be capable of CD4⁺ memory T cell activation. We are unable to formally test that however, since the antigen-specificity of intratumoral T cells is unknown.

3. The authors show no presentation by ILCs directly *ex vivo* without prolonged cytokine treatment but state in the discussion (Lines 358-360), that it could be different *in vivo* due to key differences in T cell composition, microenvironment, etc...

Response: The translational nature of this study comes with some limitations that cannot easily be overcome. We try to be as transparent as possible with these limitations. We refer to our response to the above question 2 for the *in vivo* relevance of our data.

4. Related to this, the title includes the phrase “cytokine microenvironment” but this microenvironments are not directly examined.

Response: We refer to our response to question 2, point 2.

Additional concerns:

5. Other than cytokine responsiveness and lack of co-stimulatory molecules, the peripheral and intestinal ILCs are not functionally connected. Are there data showing that they have similar transcriptional programs, for instance?

Response: Based on published data, human intestinal and blood ILCs are likely different in terms of differentiation status, blood ILCs being mostly ILCP (Lim et al., Cell, 2017) and intestinal ILCs mostly ILC3 (Bernink et al., 2013). However, in terms of antigen-presenting capacity there seem to be parallels (HLA-DR but no co-stimulatory molecule expression), which we exploited. Using blood ILCs as a model system allowed us to make use of the relative high abundance and frequency of CMV-responsiveness that can be detected in human peripheral blood donors. To understand more about the phenotype and function of blood, tumor and non-affected colon tissue we analyzed markers associated with tissue-residency and maturation of ILCs, including CD69, NKp44 and ROR γ t (supplementary figure 1G and 2A-B). The results show that whereas circulating blood ILCs are CD69⁻ (as reported by Lim et al, 2017), non-affected colon tissue ILCs are predominantly tissue resident mature ROR γ t⁺NKp44⁺ ILC3. Tumor-associated ILCs express significantly less CD69, NKp44 and ROR γ t, suggesting that they are less tissue resident and display a less pronounced ILC3-phenotype than non-affected colon tissue, thus might contain an infiltrating sub-population of circulating ILCs. Of note, the elevated HLA-DR expression on tumor ILCs is seen on both CD69⁺ and CD69⁻ cells (Supplementary Fig. 1H), as well as on NKp44⁺ and NKp44⁻ ILCs (data not shown). This is discussed on pages 6 and 19.

Although we agree with the reviewer that a transcriptional comparison of blood and gut ILCs would be very valuable, it is beyond the scope of the current paper, which consists of 7 main and 19 supplementary figures.

6. Are there conditions *in vivo* where ILCs are shown to express co-stimulatory molecules? I looked for this information in the manuscript and did not find it. If I did not miss it, this is important to mention or show. Otherwise, the observations risk being only *in vitro* phenomena.

Response: We thank the reviewer for pointing this out. There are reports on the co-stimulatory molecule expression on ILCs *in vivo*, which are based on mouse studies (von Burg et al., PNAS, 2014; Oliphant et al, Immunity, 2014; Castellanos, Immunity, 2018). We have detected low levels of CD86 expression on ILCs in tumor tissues of three CRC patients, indicating that this could happen *in vivo* in the context

of CRC. We now added this data as a new Supplementary Fig. 5. This is discussed on pages 7 and 18 of the manuscript.

7. On this note, for colonic ILCs the authors show upregulation of class II and co-stimulatory molecules but actual antigen presentation (even with peptide) is not shown. In general, the authors lead with surface markers and then follow, almost as an afterthought, with actual antigen presentation, which does not make sense to me in light of the previous points.

Response: We refer to the reviewer's question 2, in particular point 5 for this answer. In brief, examining the antigen-presenting capacity of intratumoral ILCs is not possible since the antigen-specificity of human intratumoral CD4⁺ T cells in CRC is unknown. It is also not possible to obtain the amount of blood needed from a CMV⁺ CRC patient, in order to test the capacity of intratumoral ILCs to present CMV peptide to blood CMV-specific CD4⁺ T cells.

8. The work is largely descriptive. An exception is the investigation of signaling requirements mentioned above. There are two problems with these studies (Figure 5). First, the authors do not address the possibility that reduced expression of the various proteins caused by BAY11-7082 is due to toxicity, even though toxicity to ILC2 cells was noted (lines 276-277).

Response: Indeed, BAY11-7082 is known to be cytotoxic. In ILC3-like cells, HLA-DR and co-stimulatory molecule upregulation is detectable at day 3 of IL-1 β treatment (Fig. 4A, B). The addition of BAY11-7082 for this period of time still allowed us to analyze the viable cells. Non-viable cells were excluded based on the CD45 and dead cell marker staining. Viability of BAY11-7082 treated ILC3 was on average 55% and 36% for 0,5 μ M and 1 μ M of inhibitor, respectively. We have included a representative experiment below for the reviewer's inspection only (DCM: dead cell marker). ILC2, in contrast, needed longer cytokine-culture time (5 days) to upregulate HLA-DR and co-stimulatory molecules (Fig 4C, D). Culturing ILC2 with BAY11-7082 for 5 days caused more cell death and thus, could not be analyzed.

We also show viability data for the other NFκB inhibitor added to this study - BMS-345541 (Supplementary figure 16), below for the reviewer's inspection only.

Second, the authors do not show that effective doses of Flurabine phosphate were utilized. (Is there a STAT1-dependent process in ILCs that could have served as a positive control?)

Response: We thank the reviewer for pointing this out. We attempted to address this question by stimulating ILCs with IFN- γ , which should cause STAT1 activation. Indeed, we could detect phosphorylation of STAT1-Tyr710 by IFN- γ (data shown below for the reviewer's inspection only) and, as we already reported in figure 5, STAT1-Ser737 phosphorylation by IL-1 β . Fludarabine did not influence phosphorylation of any of these sites. We also tried to stimulate ILCs with IFN- γ for three days to assess the effect of fludarabine on HLA-DR expression however the viability of ILCs cultured with the combination of IFN- γ and fludarabine was severely impacted, making it impossible to evaluate the effect. Hence, we are unsure of the effectiveness of the inhibitor fludarabine. And unfortunately, there is currently no specific STAT1 inhibitor available on the market. Due to this, we have removed the fludarabine data. This is not affecting the message of our paper, since the IL-1 β induced upregulation of HLA-DR (and co-stimulatory molecules) is unlikely to be mediated by STAT1, since this would require phosphorylation of the dimerization site STAT1-Tyr710, which is not seen with IL-1 β (figure 5). NF-kB induced HLA-DR upregulation has previously been reported by Lee *et al.*, EJI, 2006, which is mentioned in the results section on page 13.

9. For the imaging experiments in Figure 1, co-localization of class II-positive cells

and CD4 cells does not mean activation. An analysis that shows actual activation *in situ* would substantially strengthen the work.

Response: We agree with the reviewer that demonstration of T cell activation by intestinal ILCs *in situ* in humans would greatly increase the impact of this work. However, conventional T cell “activation markers” are also markers for tissue residency (e.g. CD69) or T cell exhaustion (e.g. PD1) and therefore cannot be used as readout. Taking into consideration that the cytokine response of an activated colon-resident memory T cell needs a specific antigen, the frequency of the CD4⁺ memory T cells responding to their specific stimulus at the given time point would be very low and difficult to detect using microscopy. We believe that addressing this question is not feasible in humans and is beyond the scope of this paper.

10. Related to this, the authors observe that ILCs, HLA-DR^{hi}CD45⁺ and T cells frequently co-localize, suggesting “mutual regulatory mechanisms”. (Lines 129-131). Are the authors suggesting a processing and presentation function for ILCs in this context?

Response: One possibility is that the HLA-DR^{hi}CD45⁺ cells provide the cytokines that are needed for the upregulation of HLA-DR that is seen on intestinal versus blood ILCs (figure 1 and supplementary figure 1F). The HLA-DR^{hi}CD45⁺ cells would likely be better antigen-presenting cells, but they might also be recruited to lymph nodes upon activation, possibly rendering unprofessional APCs, such as ILCs, important as inducers of CD4⁺ T cell recall responses in the intestine. The potential mutual regulatory mechanisms between ILCs and CD45⁺HLA-DR^{hi} cells are discussed on page 21.

11. Inhibitor experiments were carried out following cytokine exposure for 1 hour at most when other experiments were carried out after several days of cytokine exposure, shorter incubations not leading to functional differences. Is the several-days exposure to the three cytokines realistic? Could this happen *in vivo*?

Response: The short-term experiments that the reviewer is referring to are likely the phosflow data presented in figure 5. Phosphorylations are early activation events that happen within minutes and that is the reason for culturing the cells for only 10, 30 and 60 minutes. The NFκB inhibitors that were used (figure 5 and supplementary figure 16) affect mRNA and protein levels of NFκB. Hence, to see an effect of this we had to culture the cells for longer (three days), to be able to assess the effects of reduced NFκB protein. Several days of exposure to cytokines are relevant in the setting of chronicity, which is the case for the tumor environment, which develops during a time span of years.

12. Figure 7: The data for no treatment (“Ex vivo”) are not provided for the tumor border and tumor.

Response: The *ex vivo* data is shown in figure 1 (HLA-DR for all intestinal regions) and supplementary figure 5 (co-stimulatory molecules for all intestinal regions).

13. The p values in Figure 1B are really 0.003, 0.005 and 0.002?

Response: Yes. We used a paired, non-parametrical statistical test. Although the inter-donor variability is high, the intra-donor changes are very robust.

Reviewer #3 (Remarks to the Author):

In this manuscript, Rao et al. investigate the antigen presenting capacities of human ILCs and how cytokines regulate this function in vitro. They show that ILCs present in colorectal tumors (or their vicinity) express elevated levels of HLA-DR although lacking co-stimulatory molecules on their surface. Blood ILCs were able to uptake and process ovalbumin and when stimulated by IL-1b demonstrate features of antigen presenting cells. These cells were subsequently able to induce recall responses in memory T cells. Overall, the authors’ main conclusions are supported by their presented data which appear to be mostly novel. Experiments are well-conducted. This manuscript confirms and complements the findings of Ohne et al (NatImmunol 2016), describing a role for IL-1b in ILC2 maturation and function. However, there are a few concerns with the manuscript in its current form as follows:

1. Authors should cite the paper of Ohne et al (NatImmunol 2016), since it is complementary to their study, especially because they are showing cytokines production and transcriptome changes in ILC2 stimulated with IL-1b.

Response: We thank the reviewer for pointing out that we missed to cite this relevant work. We now have done so in the results (page 12) and the discussion (page 19) sections.

2. Throughout the manuscript, the authors state that both IL-1b and IL-18 induce the antigen-presenting phenotype observed in ILCs. The data don’t support this conclusion as only IL-1b is inducing upregulation of MHCII and costimulatory molecules. IL-18 alone doesn’t seem to have any role and doesn’t show any additive or synergistic effect in combination with IL-1b except slightly on ILC2s (Figures 4, 5, 6). It is therefore misleading to state that IL-18 has a role in this ILC function since it doesn’t seem to have a role on ILC3 and a weak role on ILC2. The authors should

correct this in the manuscript and be less affirmative on the role of IL-18 (lines 240, 284 for example).

Response: We thank the reviewer for this remark. In fact, 5-day treatment with IL-18 does have a significant effect on HLA-DR, CD80, CD70 and CD86 expression on ILC2 and ILC3-like cells (Fig 4A). The magnitude of the upregulation is indeed drastically lower than that induced by IL-1 β , which is probably also the reason for the absent additive effect.

Nonetheless, since we do not present the data on the actual T cell stimulation by ILCs expanded with IL-2 and IL-18, we have now toned down the statements on the significance of IL-18 (Abstract: page 2, Introduction: page 4-5, Results: page 14)

3. In the abstract, line 28, « intestinal and peripheral blood ILCs... » is also misleading, since only blood ILCs were actually tested ex vivo for APC characteristics in several cytokine conditions. Only Figure 1 shows intestinal ILCs.

Response: We agree with the reviewer and have now changed the abstract.

4. In Figure 1A, the CD117⁺ and CD117⁻ subsets are difficult to distinguish. The authors should modify the y axis so the two populations become distinct. This should be feasible with analyzing softwares such as diva or flowjo or by adding a cross on the dot plot.

Response: We have now adjusted the y-axis so that CD117⁺ and CD117⁻ events are better separated.

5. Concerning figure 1C, D, the authors observe that ILC co-localize with T cells, with APCs or both (« frequently observed » line 129). It would be interesting and would add to the manuscript if the authors could quantify the different interactions in the several anatomical regions. If ILCs are more often observed in contact with a T cell or with Tcell+APC than alone, that would strengthen the proposed hypothesis that ILC can present antigens and reactivate memory T cells.

Response: We thank the reviewer for this suggestion. We now followed the reviewer's recommendation and present the results in the figure 1E.

6. In figure 2C, D, the authors do not show data on ILC2, even though Fig 2A shows ILC2. This information on whether ILC2 are able to process antigen (DQ-OVA here) is crucially missing, especially regarding the conclusions of the manuscript emphasizing on the potential of ILC2 to act as antigen-presenting cells. Also, as it is now, the

manuscript states that « degradation is equally efficient in all ILC subsets » (line 158) which is untrue since data on ILC2 is missing.

Response: We thank the reviewer for this point. We have now included the information on the DQ-OVA uptake and degradation by ILC2 in the Fig. 2C and D.

7. In figure 4, the authors should show upregulation of HLA and costimulatory molecules on professional APCs such as Lin+HLA-DRhi cells (as in Figure 2) in parallel to the data on ILC3 and ILC2. This would greatly help to evaluate the extend of the upregulation of these molecules and the potential « APCness » of ILCs. Figure 2 clearly demonstrates that even though ILCs seem to be able to act as APCs, they are way less efficient at processing DQ-OVA than professional APCs. The parallel should be made for levels of HLA and costim molecules as well.

Response: We thank the reviewer for this comment. To understand the “APC-ness” of stimulated ILCs we compared HLA-DR and co-stimulatory molecule expression to *ex vivo* isolated monocytes/DCs. Expression of HLA-DR and co-stimulatory molecules on unstimulated bulk professional APCs from peripheral blood is shown in Fig. 3D and Supplementary Fig. 5. Stimulated ILC3-like cells upregulate HLA-DR and CD86 to levels comparable to unstimulated monocytes/DCs. This is now specifically mentioned on page 11.

8. Figure 4 shows that ILC2 respond to IL-23. Are there other studies that describe ILC2 responding to IL-23 ? if so, authors should provide references. Isn't it possible that these cells are ILC2 that converted to an ILC3 phenotype? The authors mention that these stimulated ILC2 and ILC3like cells do not express NKG2A or CD94, ruling out that these cells acquire an ILC1/NK-like phenotype. However, supp Figure 10 shows a strong upregulation of NKp44 on ILC3 cells after cytokine stimulation, that could suggest that these cells are converting to an NCR+ ILC3 or ILC1-like phenotype. Ohne et al. (NatImmuno 2016) show that IL-1b-stimulated ILC2 upregulate Tbet, Eomes and several cytokine receptors and differentiate into ILC1-like cells. How do the authors interpret their results in view of the Ohne study ? Staining for Tbet, Eomes, RORgt for exemple would add greatly to the manuscript and to understand the complex role of cytokines in ILC function and plasticity.

Response: This question is similar to question 9 raised by reviewer 1 so we also refer to our answer to that question. As suggested by the reviewer we have analyzed the transcription factor expression of IL-1 β -stimulated ILC3-like cells and ILC2 (supplementary Fig. 13D). We observed that both ILC2 and ILC3-like cells stimulated with IL-1 β display a high expression level of ROR γ t. Although ILC3-like cells upregulated GATA3, IL-1 β stimulated ILC2 displayed the highest level of this

transcription factor. Interestingly (and in line with the observation by Ohne et al.) IL-1 β stimulated ILC2 showed upregulation of T-bet. We could not detect Eomes expression on any of the ILC subsets on the protein level.

There are indeed several studies that have unveiled that IL-1 β is a necessary co-factor driving ILC plasticity (Bal et al, Nat Immunol, 2016; Silver et al., Nat Immunol, 2016; Ohne et al., Nat Immunol, 2016; Golebski et al., Nat Comm, 2019; Bernink et al, Nat Immunol, 2019; Hochdörfer et al, EJI, 2019; Mazzurana et al., EJI, 2019).

Importantly, these studies show that IL-1 β alone is not sufficient for full differentiation to other ILC lineages but that this requires polarizing cytokines such as IL-12 (for ILC1; Bernink et al, Nat Immunol, 2013; Mazzurana et al., EJI, 2019), IL-23 and TGF- β (for ILC3; ; Golebski et al., Nat Comm, 2019; Bernink et al, Nat Immunol, 2019; Hochdörfer et al, EJI, 2019).

We interpret these results as IL-1 β is acting as a “gate opener”, while additional second signal is necessary to drive the final lineage commitment, which is fully in line with the study of Ohne et al. We have now discussed this observation on pages 12 and 19.

9. The conclusion stated on line 284 that IL-1b and IL-18 stimulation leads to NF κ b signaling needed for optimal HLA and costimulatory molecule expression is not supported by the data in the figure 5 which doesn't show HLA and costimulatory molecule expression on cells stimulated by IL-18 (Figure 5C only shows data for IL-1b). The authors should correct this.

Response: We thank the reviewer for this comment. We have now corrected this.

Editorial Note: The original Reviewers 1 and 3 were unavailable to comment on the revisions. To evaluate author response to the points raised by the original Reviewers 1 and 3, a new Reviewer 3 was recruited in this round.

REVIEWERS' COMMENTS:

Reviewer #2 (Remarks to the Author):

The authors have been quite responsive to the reviews. I continue to have reservations about the conclusions drawn but understand the limitations of the system. One remaining issue:

The data in the new Figure 1E in response to Reviewer 1's important concern (#6 in the rebuttal) should be analyzed for statistical significance.

Reviewer #3 (Remarks to the Author):

The authors answered all remaining questions and responded to all comments. This manuscript robustly presents new data and confirms already published data. It is ready for publication in its current form.

Dr. Mathilde Girard-Madoux

Remaining reviewer comments:

Reviewer #2 (Remarks to the Author):

The data in the new Figure 1E in response to Reviewer 1's important concern (#6 in the rebuttal) should be analyzed for statistical significance.

Response: Since N (donors) = 3, we cannot analyze the statistical significance using the non-parametric test, due to low sample number. Therefore, we have removed the comparative statement about the colocalization of ILCs in non-affected and tumor tissue.